# *Polycomb* regulates circadian rhythms in *Drosophila* in clock neurons

Xianguo Zhao[1,*], Xingzhuo Yang[1,*], Pengfei Lv[1,*], Yuetong Xu[2], Xiangfeng Wang[2], Zhangwu Zhao[1], Juan Du[1]

**Circadian rhythms are essential physiological feature for most living organisms. Previous studies have shown that epigenetic regulation plays a crucial role. There is a knowledge gap in the chromatin state of some key clock neuron clusters. In this study, we show that circadian rhythm is affected by the epigenetic regulator *Polycomb* (*Pc*) within the *Drosophila* clock neurons. To investigate the molecular mechanisms underlying the roles of *Pc* in these clock neuron clusters, we use targeted DamID (TaDa) to identify genes significantly bound by Pc in the neurons marked by *C929-Gal4* (including l-LNvs cluster), *R6-Gal4* (including s-LNvs cluster), *R18H11-Gal4* (including DN1 cluster), and *DVpdf-Gal4*, *pdf-Gal80* (including LNds cluster). It shows that Pc binds to the genes involved in the circadian rhythm pathways, arguing a direct role for *Pc* in regulating circadian rhythms through specific clock genes. This study shows the identification of Pc targets in the clock neuron clusters, providing potential resource for understanding the regulatory mechanisms of circadian rhythms by the PcG complex. Thus, this study provided an example for epigenetic regulation of adult behavior.**

## Introduction

The circadian rhythm in the *Drosophila* brain is executed by 150 clock neurons including three groups of lateral neurons, s-LNvs (small ventral–lateral neurons), l-LNvs (large ventral–lateral neurons), and LNds (dorsal–lateral clock neurons), and three groups of dorsal neurons, DN1s (dorsal neurons 1), DN2s (dorsal neurons 2), and DN3s (dorsal neurons 3). Both s-LNvs and l-LNvs are PDF-releasing cells and only s-LNvs have the PDFR receptor (Shafer et al, 2008; Im & Taghert, 2010; Kula-Eversole et al, 2010; Im et al, 2011; Klose et al, 2016). These two groups of neurons however have quite different functions. It was shown in previous studies that l-LNvs respond to the light signal by releasing the neuropeptide PDF (pigment-dispersing factor) (Rieger et al, 2006). Furthermore,

s-LNvs act as major pacemakers, where the loss of s-LNvs, also referred to as morning neurons, results in a lack of morning activity (Grima et al, 2004; Stoleru et al, 2005). So, different clusters of clock neurons play different roles in circadian regulation. Understanding the gene expression features of different clusters of clock neurons is important for dissecting the regulatory mechanism of circadian rhythm.

RNA sequencing of different groups of clock neurons was carried out to identify the neuronal cluster-specific expressed genes (Kula-Eversole et al, 2010; Nagoshi et al, 2010). Nevertheless, there is a knowledge gap in understanding the chromatin states of these neuronal clusters and their relationship with their functions. Closing this gap helps to understand the genetic regulatory network in diversified neuronal clusters.

Chromatin is composed of DNA and all the associated proteins including histones and other chromatin-binding proteins. It is also essential for transcriptional regulation and other processes. Histone modifications and the chromatin protein-binding profile interact and form a major determinant of the genes' transcriptional activities. According to the protein-binding profiles, five types of chromatin states were identified (Filion et al, 2010). The chromatin states include two types of heterochromatins defined by, for example, Pc (Polycomb) and HP1 (heterochromatin protein 1), one type of repressive chromatin defined by, for example, histone H1, SUUR (suppressor of under-replication), and two types of transcriptionally active euchromatin marked by, for example, BRM (brahma) and MRG15 (MORF-related gene 15). Pc protein is a component of PcG complex and has been proved important both in development and disease (Sawarkar & Paro, 2010). E(Z) (enhancer of zeste), which is another component of PcG complex, has been proved to be important in circadian regulation (Etchegaray et al, 2006). However, the downstream mechanism needs further study.

Furthermore, efforts have been made to identify the global chromatin states of different types of cells. Epigenomic landscapes of different types of primary tissues/cells in humans and other modal organisms were also identified (Landt et al, 2012; Roadmap Epigenomics Consortium et al, 2015). Nevertheless, to the best of our knowledge, the chromatin states of more specific tissues in vivo

[1]Department of Entomology and MOA Key Lab of Pest Monitoring and Green Management, College of Plant Protection, China Agricultural University, Beijing, China  [2]Department of Crop Genomics and Bioinformatics, College of Agronomy and Biotechnology, National Maize Improvement Center of China, China Agricultural University, Beijing, China

Correspondence: dujuan9981@cau.edu.cn
*Xianguo Zhao, Xingzhuo Yang, and Pengfei Lv contributed equally to this work

have not been identified. This has mainly been because of technical issues where the initial physiological states of the cells were damaged in the process of isolating specific tissue/cells. To address this issue, in vivo-targeted DamID (TaDa) was established (Southall et al, 2013; Pindyurin et al, 2016; La Fortezza et al, 2018), which enables precise identification of the epigenomic landscapes of tissues or cells in vivo, hence better identification of chromatin states.

In this study, we applied the tissue-specific TaDa method (Southall et al, 2013; Marshall & Brand, 2015) to identify the Polycomb (Pc) binding profile of four groups of neurons. The considered neurons marked by C929-Gal4 (including l-LNvs cluster neurons) (Taghert et al, 2001; Grima et al, 2004), R6-Gal4 (including s-LNvs cluster neurons) (Helfrich-Förster et al, 2007), R18H11-Gal4 (including DN1 cluster neurons) (Guo et al, 2016), and DVpdf-Gal4, pdf-Gal80 (including LNds cluster neurons) (Schubert et al, 2018). Comparing the Pc binding profiles of these groups of neurons, we then identified the specific features of the Pc binding profile within different clock neuron clusters. Remarkably, the functional study of Pc protein in different clock neuron clusters indicated that Pc regulates circadian rhythm mainly in ventral lateral neurons (l-LNvs and s-LNvs). Analysis of the Pc binding profile also revealed that Pc regulates multiple clock genes and clock-regulated genes in these clock neuron clusters. Based on the above, we then concluded that in Drosophila, Pc regulates circadian rhythms in clock neurons.

# Results

### Polycomb (Pc) is required for regulating the circadian rhythm of clock neurons

To examine the role of Pc in the circadian rhythm regulation in Drosophila melanogaster, we studied the effect of down-regulation of Pc in the clock neurons that express tim-Gal4. For consistency, we used male Drosophila in this study. In the constant darkness (DD) conditions, the tim-Gal4/PcRNAi flies showed a severely disrupted circadian rhythm (Table 1). Circadian rhythm can be measured by percentage rhythmicity (the percentage of rhythmic animals) and the power value (indicating the stability of the circadian oscillation). In contrast to the $w^{1118}$ and two control group flies including UAS-PcRNAi and tim-Gal4, both of which are 100% in percentage rhythmicity, 61.3% of the tim-Gal4/UAS-PcRNAi flies were rhythmic (Table 1). The activity counts measure the average of the peak activity counts in 30 min. Our results indicated that tim-Gal4/UAS-PcRNAi flies had a significantly reduced activity counts profile during the night, but not during the day (Figs 1A–D″, G, and H; Table 1). The power value was also reduced in tim-Gal4/UAS-PcRNAi flies (Fig 1I; Table 1). The period length was not significantly changed (Table 1). Nevertheless, the up-regulation of Pc (tim-Gal4/UAS-Pc) in tim-expressing neurons did not cause a significant circadian phenotype (Fig S1A–C″ and O; Table 1). The above observations indicate that the Pc loss in tim-expressing neurons affects the circadian rhythm.

To further explore which of the clock neurons are required for the Pc gene to regulate the circadian rhythm, we down-regulated Pc

in the pdf-Gal4–marked (including l-LNvs and s-LNvs cluster), R18H11-Gal4–marked (including DN1s cluster), and DVpdf-Gal4, pdf-Gal80–marked (including LNds cluster) neurons. It was seen that the circadian rhythm effects of pdf-Gal4/UAS-PcRNAi flies partially represented the phenotypes of the tim-Gal4/UAS-PcRNAi flies. Compared with the controls, the rhythmic percentage of pdf-Gal4/UAS-PcRNAi flies is 83.9% in the DD condition, indicating a circadian rhythm defect (Table 1). Furthermore, pdf-Gal4/UAS-PcRNAi flies were verified to have a significantly reduced activity counts profile during the night, but not during the day (Fig 1A–B″, E–F″, G, and H; Table 1). The power value in rhythmic pdf-Gal4/UAS-PcRNAi flies was also significantly reduced (Fig 1A–B″, E–F″, and I; Table 1). Nevertheless, PcRNAi driven by the other two clock neuron Gal4s had no significant phenotypes in the rhythmic percentage and power value (Fig S1D–H″, and O; Table 1). The activity count was significantly reduced in R18H11-Gal4/UAS-PcRNAi during the night and was significantly increased in DVpdf-Gal4, pdf-Gal80/UAS-PcRNAi flies during the day (Fig S1M and N; Table 1).

The role of Polycomb in development has been well established. To eliminate the developmental effects of Pc, we did experiments with UAS-Dicer to enhance the RNAi effects and $Gal80^{ts}$ to temporally induce Pc RNAi in the adult stage. The results showed that both of the tim-Gal4– and the pdf-Gal4–driven Pc RNAi had reduction in percentage rhythmicity compared with the controls (Table 2), whereas the Dvpdf-Gal4, pdf-Gal80, and R18H11-Gal4–driven Pc RNAi had no significant changes compared with the controls, which was consistent with the results in Table 1. In conclusion, Pc was required for the circadian rhythm in clock neurons of adult flies.

To elucidate if the activity defects were because of the locomotion deficits or circadian disruption, we calculated the activity per minute of the experimental and control genotypes. We found that the all the genotypes except Dicer; pdf-Gal4/UAS-Pc RNAi (which had mild but significant reduction) had no significant changes in activity per minute (Fig S2A). Further climbing test found that all the genotypes had no significant changes in activity performance indicated by the climbing speed and % climbing success (Fig S2B and C). So, we concluded that the activity defects we see in this study were because of the circadian disruption rather than locomotion deficits.

We further up-regulated Pc in the three clusters of clock neurons including marked by pdf-Gal4 (including LNvs and s-LNvs cluster), R18H11-Gal4 (including DN1s cluster), and DVpdf-Gal4, pdf-Gal80 (including LNds cluster). The results showed no significant circadian rhythmic phenotypes in rhythmic percentage and power value between the controls and Pc overexpression flies (Fig S1I–L″, and O; Table 1). Nonetheless, the up-regulation of Pc significantly reduced the activity counts during the night in neurons marked by pdf-Gal4 (including l-LNvs and s-LNvs cluster) and DVpdf-Gal4, pdf-Gal80 (including LNds cluster) and during the day and night in neurons marked by R18H11-Gal4 (including DN1s cluster) (Fig S1I–L″, M, and N; Table 1).

Phase shift experiments indicated that Pc was required for proper response to environmental changes. We challenged the flies by delaying the light phase for 8 h. Calculation of the percentage of flies that had phase shift more or less than 8 h indicated that there was an increase of the fly percentage that have phase shift less

**Table 1.  Locomotor activity of flies with altered *Pc* levels under darkness condition.**

| Genotype | Rhythmic (%) | Period ± SEM | Power ± SEM | Activity ± SEM | N |
|---|---|---|---|---|---|
| Control phenotypes | | | | | |
| *w^1118* | 100 | 24.0 ± 0.02 | 97.1 ± 5.51 | 24.07 ± 1.53 | 32 |
| *pdf-Gal4/+* | 100 | 24.0 ± 0.05 | 111.5 ± 3.36 | 33.77 ± 1.44 | 32 |
| *tim-Gal4/+* | 100 | 24.0 ± 0.03 | 119.3 ± 5.66 | 32.13 ± 1.59 | 31 |
| *R18H11-Gal4/+* | 100 | 23.8 ± 0.05 | 111.5 ± 3.80 | 37.10 ± 1.83 | 32 |
| *Dvpdf-Gal4, pdf-Gal80/+* | 96.7 | 24.2 ± 0.05 | 96.3 ± 3.47 | 33.84 ± 1.17 | 30 |
| *UAS-PcRNAi/+* | 100 | 23.9 ± 0.08 | 108.0 ± 4.46 | 32.93 ± 1.46 | 31 |
| *UAS-Pc/+* | 96.8 | 24.0 ± 0.03 | 89.6 ± 5.57 | 29.14 ± 1.39 | 31 |
| *Pc* down-regulation phenotypes | | | | | |
| *pdf-Gal4/UAS-PcRNAi* | 83.9**** | 23.8 ± 0.07 n.s. | 68.4 ± 3.62**** | 15.31 ± 0.70**** | 31 |
| *tim-Gal4/+; UAS-PcRNAi/+* | 61.3**** | 24.0 ± 0.03 n.s. | 50.6 ± 3.72**** | 18.69 ± 1.57**** | 31 |
| *R18H11-Gal4/UAS-PcRNAi* | 100 n.s. | 24.0 ± 0.04** | 101.5 ± 3.07 n.s. | 24.11 ± 1.82**** | 40 |
| *Dvpdf-Gal4, pdf-G80/+; UAS-PcRNAi/+* | 96.9 n.s. | 24.1 ± 0.04** | 82.3 ± 4.92*** | 31.23 ± 1.64 n.s. | 32 |
| *Pc* up-regulation phenotypes | | | | | |
| *UAS-Pc/+; pdf-Gal4/+* | 100 n.s. | 23.9 ± 0.04**** | 123.5 ± 4.32**** | 16.79 ± 1.18**** | 32 |
| *UAS-Pc/tim-Gal4* | 96.8 n.s. | 24.2 ± 0.05**** | 117.2 ± 4.24*** | 34.62 ± 1.49 * | 31 |
| *UAS-Pc/+; R18H11-Gal4/+* | 93.8 n.s. | 24.0 ± 0.00 n.s. | 98.3 ± 5.76** | 20.32 ± 0.87**** | 32 |
| *UAS-Pc/Dvpdf-Gal4, pdf-Gal80* | 100 n.s. | 24.1 ± 0.03**** | 82.7 ± 4.32 n.s. | 23.27 ± 1.21**** | 32 |

Activity shows the average of the peak activity counts in 30 min of all files during its 24-h activity cycle. Power shows the strength of the circadian oscillation of the rhythmic flies during DD, and individual flies with a power (≥10) and a "width" value of 1.5 or more (denotes number of peaks in 30 min increments above the periodogram 95% confidence line) were considered rhythmic. Statistical differences were measured using Fisher's exact test for %Rhythmic; n.s. indicates no significant difference. ****$P < 0.0001$. Statistical differences were measured using one-way ANOVA for period, activity, and power; n.s. indicates no significant difference, * indicates $P < 0.05$, ** indicates $P < 0.01$, *** indicates $P < 0.001$, **** indicates $P < 0.0001$. As for statistical analysis of **Period**, ANOVA $F_{[2, 86]} = 2.168$, $P = 0.121$ for *pdf-Gal4/UAS-PcRNAi*; ANOVA $F_{[2, 78]} = 2.316$, $P = 0.105$ for *tim-Gal4/+; UAS-PcRNAi/+*; ANOVA $F_{[2, 100]} = 5.039$ $P = 0.008$ for *R18H11-Gal4/UAS-Pc RNAi*; ANOVA $F_{[2, 88]} = 6.186$, $P = 0.003$ for *Dvpdf-Gal4, pdf-G80/+; UAS-PcRNAi/+*; ANOVA $F_{[2, 91]} = 2.445$, $P = 0.092$ for *UAS-Pc/+; pdf-Gal4/+*; ANOVA $F_{[2, 88]} = 10.752$, $P = 6.64 \times 10^{-5}$ for *UAS-Pc/tim-Gal4*; ANOVA $F_{[2, 88]} = 14.829$, $P = 2.82 \times 10^{-6}$ for *UAS-Pc/+; R18H11-Gal4/+*; ANOVA $F_{[2, 88]} = 8.654$, $P = 3.71 \times 10^{-4}$ for *UAS-Pc/Dvpdf-Gal4, pdf-G80*. As for statistical analysis of **activity**, ANOVA $F_{[2, 91]} = 68.547$, $P = 6.95 \times 10^{-19}$ for *pdf-Gal4/UAS-PcRNAi*; ANOVA $F_{[2, 90]} = 18.897$, $P = 1.41 \times 10^{-7}$ for *tim-Gal4/+; UAS-PcRNAi/+*; ANOVA $F_{[2, 100]} = 15.299$, $P = 1.60 \times 10^{-6}$ for *R18H11-Gal4/UAS-Pc RNAi*; ANOVA $F_{[2, 90]} = 0.844$, $P = 0.433$ for *Dvpdf-Gal4, pdf-G80/+; UAS-PcRNAi/+*; ANOVA $F_{[2, 92]} = 43.275$, $P = 5.67 \times 10^{-14}$ for *UAS-Pc/+; pdf-Gal4/+*; ANOVA $F_{[2, 90]} = 3.375$, $P = 0.039$ for *UAS-Pc/tim-Gal4*; ANOVA $F_{[2, 92]} = 35.364$, $P = 4.04 \times 10^{-12}$ for *UAS-Pc/+; R18H11-Gal4/+*; ANOVA $F_{[2, 90]} = 17.724$, $P = 3.23 \times 10^{-7}$ for *UAS-Pc/Dvpdf-Gal4, pdf-G80*. As for statistical analysis of **power**, ANOVA $F_{[2, 91]} = 45.572$, $P = 1.93 \times 10^{-14}$ for *pdf-Gal4/UAS-PcRNAi*; ANOVA $F_{[2, 90]} = 72.843$, $P = 1.53 \times 10^{-19}$ for *tim-Gal4/+; UAS-PcRNAi/+*; ANOVA $F_{[2, 100]} = 1.953$, $P = 0.147$ for *R18H11-Gal4/UAS-PcRNAi*; ANOVA $F_{[2, 90]} = 8.601$, $P = 3.82 \times 10^{-4}$ for *Dvpdf-Gal4, pdf-G80/+; UAS-PcRNAi/+*; ANOVA $F_{[2, 92]} = 15.843$, $P = 1.22 \times 10^{-6}$ for *UAS-Pc/+; pdf-Gal4/+*; ANOVA $F_{[2, 90]} = 9.207$, $P = 2.30 \times 10^{-4}$ for *UAS-Pc/tim-Gal4*; ANOVA $F_{[2, 92]} = 5.393$, $P = 0.006$ for *UAS-Pc/+; R18H11-Gal4/+*; ANOVA $F_{[2, 90]} = 1.102$, $P = 0.337$ for *UAS-Pc/Dvpdf-Gal4, pdf-G80*.

than 8 h (Fig 1J and K), indicating a defect in light entrainment in the *Pc* RNAi.

The above observations collectively indicate that in clock neurons, normal levels of *Pc* are required for standard circadian activity, but that overexpressing *Pc* is not sufficient to disrupt those circadian patterns. Furthermore, *Pc* activity was required for proper light entrainment.

### Genome-wide DNA-binding profile of Pc in different clusters of clock neurons

To investigate Pc binding profiles of specific neurons within the *Drosophila* circadian circuit, we adapted the TaDa method. This method was used in the previous gene expression analyses of *Drosophila* brain neurons (Korzelius et al, 2014; Zhao et al, 2021b). The TaDa method is based on DamID, a technology to widely detect binding profiles of proteins on the DNA genome. By coupling this method with the Gal4 system in *Drosophila*, both temporal and spatial resolutions are achieved. TaDa ensures very low-level expression of the Dam-fusion protein and avoids potential toxicity and artifacts from overexpression (Marshall et al, 2016). We profiled different clusters of clock neurons including l-LNvs, s-LNvs, DN1s, and LNds to identify Pc binding genes enriched in the four considered clusters of clock neurons eliminating the general background bindings (Fig 2A). The "gene enrichment signatures" are then used for the different clusters of clock neurons for further molecular characterization.

We did this experiment on the four clock neuron clusters from the adult brain marked by *R6-Gal4* (including s-LNvs cluster neurons), *C929-Gal4* (including l-LNvs cluster neurons), *R18H11-Gal4* (including DN1s cluster neurons), and *DVpdf-Gal4, pdf-Gal80* (including LNds cluster neurons) (Fig 2A). The details of the experimental protocol were described by Marshall et al (2016). The *tub-Gal80^ts* was added to the UAS-DamPc/Dam to specifically induce the transgene in adult stage. The collections were made at ZT12 (zeitgeber time 12, the time

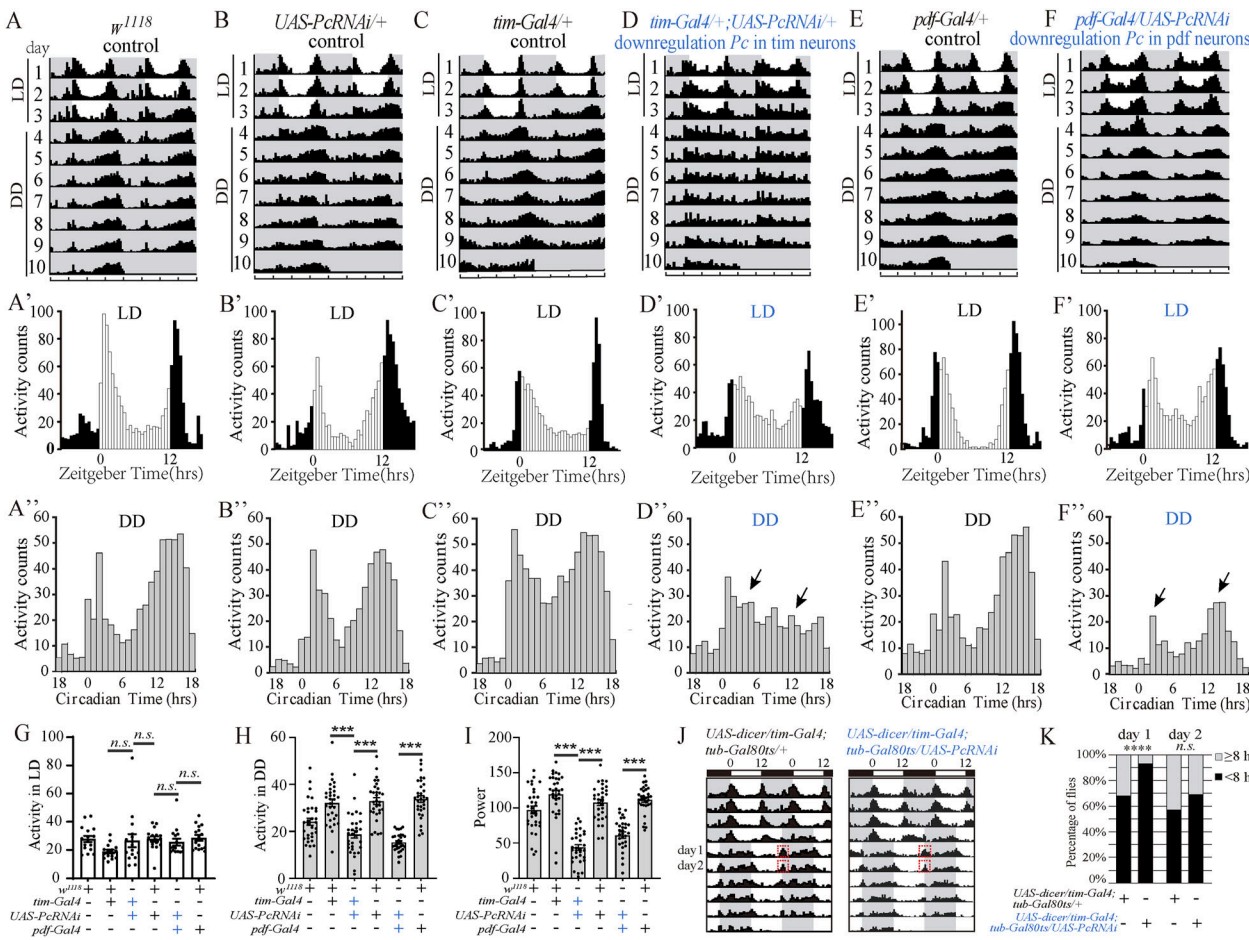

**Figure 1. *Pc* gene expression is required for behavioral circadian rhythms.**
**(A, B, C, D, E, F)** Double-plotted actograms (activity of two consecutive days shown in one row) of average locomotor activity. The flies were entrained in a 3-d 12-h light/12-h dark cycle and released to constant darkness for 7 d. The white and grey parts indicate light and dark, respectively. Actograms of 16 flies were shown for each genotype. **(A′, B′, C′, D′, E′, F′)** Locomotor activity of indicated fly strains measured during three LD days, y-axis was average activity counts in every 30 min. The white and black bars indicate day and night, respectively. **(A″, B″, C″, D″, E″, F″)** Locomotor activity of indicated fly strains measured during seven days of the constant darkness (DD), y-axis was the average activity counts in every 60 min. **(G)** Average activity counts data (30 min bins) under LD conditions. **(H)** Average activity counts data (30 min bins) under DD conditions. **(I)** Power value of the indicated genotypes, which is a measure of rhythm amplitude and corresponds to the height of the periodogram peak above the significance line. **(J)** Representative double plot actograms of respective genotypes show average locomotor activity of the indicated genotypes plotted. 4 d of 12-h light/12-h dark cycles (at constant 25°C) were followed by a delay of 8 h for the time of light off. White background color indicates light, and black background colors indicate darkness, respectively. **(K)** The phase shift percentage of different genotypes. Data information: Bar graphs are presented as mean ± SEM. **(G, H, I)** Dots on the bars are the number of tested flies, and statistical differences were measured using one-way ANOVA, with Tukey's multiple comparison test; n.s. indicates no significant difference. ***$P < 0.001$. **(K)** Statistical differences were measured using Fisher's exact test, n.s. indicates no significant difference. ****$P < 0.0001$. The genotypes of the knockout flies are marked in blue.

of lights-off under a 12-h light:12-h dark cycle) because several known cycling clock genes are at their extreme expression levels at that time point (Patke et al, 2020). We also sequenced amplicons derived from methylated genomic sequences for Dam-Pc and Dam control samples and processed the data using the damID-seq pipeline (Marshall & Brand, 2015). The generated normalized and aligned read count plots were then used to assess the level of correlation between the samples. It was found that Dam-Pc and Dam control samples cluster separately (Fig S3A). Using principal component analysis, it was also seen that the control Dam samples cluster separately from Dam-Pc samples (Fig S3B). These results indicate the Dam-Pc fusions produce specific and distinct methylation patterns compared with Dam control fusions.

To identify Pc binding genes enriched in the above four clock neurons, TaDa data were analyzed from these clock neurons. We found that Pc binds to 1,201 gene sites in *R18H11-Gal4*–expressing neurons, 2,057 gene sites in *DVpdf-Gal4*, *pdf-Gal80*–expressing neurons, 2,630 gene sites in *C929-Gal4*–expressing neurons, and 2,627 gene sites in *R6-Gal4*–expressing neurons (Table S1). Analysis of the global features of Pc binding revealed that Pc predominantly bound to the promoter region (83.4% in *C929-Gal4*, 86.73% in *DVpdf-Gal4*, *pdf-Gal80*, 84.83% in *R18H11-Gal4*, and 83.89% in *R6-Gal4*–expressing neurons, Fig 2B–E). The second most frequent binding region is the intron (Fig 2B–E). Analysis of the distribution of binding sites relative to transcriptional start sites (TSS) also revealed that the most binding sites of Pc were 1 kb around TSS (Fig 2F). The

**Table 2. Locomotor activity analysis of flies with _Pc_ down-regulation in adulthood.**

| Genotype | Rhythmic (%) | Period ± SEM | Power ± SEM | Activity ± SEM | N |
|---|---|---|---|---|---|
| Control phenotypes | | | | | |
| _UAS-dicer/tim-Gal4; tub-Gal80ts/+_ | 100 | 23.9 ± 0.13 | 71.8 ± 4.23 | 19.7 ± 1.98 | 32 |
| _UAS-dicer/tub-Gal80ts; pdf-Gal4/+_ | 90.6 | 23.7 ± 0.06 | 77.5 ± 6.46 | 16.8 ± 1.22 | 32 |
| _UAS-dicer/tub-Gal80ts; R18H11-Gal4/+_ | 59.4 | 23.8 ± 0.07 | 70.4 ± 7.76 | 24.4 ± 2.17 | 32 |
| _UAS-dicer/Dvpdf-Gal4, pdf-Gal80; tub-Gal80ts/+_ | 65.6 | 26.0 ± 0.34 | 40.0 ± 6.30 | 15.8 ± 0.84 | 32 |
| _Pc_ down-regulation phenotypes | | | | | |
| _UAS-dicer/tim-Gal4; UAS-PcRNAi/tub-Gal80ts_ | 59.1**** | 24.3 ± 0.12 n.s. | 69.8 ± 5.23 n.s. | 18.5 ± 1.30 n.s. | 44 |
| _UAS-dicer/tub-Gal80ts; UAS-PcRNAi/pdf-Gal4_ | 79.5* | 23.7 ± 0.08 n.s. | 57.1 ± 5.99* | 15.6 ± 0.99 n.s. | 39 |
| _UAS-dicer/tub-Gal80ts; UAS-PcRNAi/R18H11-Gal4_ | 54.3 n.s. | 23.9 ± 0.05 n.s. | 61.3 ± 7.73 n.s. | 19.0 ± 1.62 n.s. | 35 |
| _UAS-dicer/Dvpdf-Gal4, pdf-Gal80; UAS-PcRNAi/tub-Gal80ts_ | 59.3 n.s. | 24.8 ± 0.36* | 30.1 ± 3.39 n.s. | 21.3 ± 1.95** | 27 |

Statistical differences were measured using Fisher's exact test for %Rhythmic; n.s. indicates no significant difference; *$P < 0.05$, ****$P < 0.0001$. Statistical differences were measured using unpaired $t$ test for period, activity, and power. n.s. indicates no significant difference, * indicates $P < 0.05$, ** indicates $P < 0.01$.

binding was also decreased with increasing distance from the TSS (Fig 2G). Dam control fusion samples also showed a stronger preference for TSS compared with the Dam-Pc fusion samples (Fig 2G). In the upstream sequences proximal to the TSS of protein-coding genes, the peaks count frequency signal profiles also showed increased average enrichment scores (Fig 2H).

According to the gene peak ratios along gene distance, Pc binding genes in the four clock neurons were clustered into three classes. Cluster I consists of binding sites distributed at 3 kbp upstream and 3 kbp downstream of the TSS. Cluster II consists of binding sites concentrated at the TSS and Cluster III includes binding sites far away from the TSS (Fig 2I). Further analysis showed that the binding genes of Pc in the four clock neurons were both specific and overlapping (Fig 2J). These results indicated that the ratio of these three clusters varies in different groups of clock neurons. The above results collectively suggested the diversity of _Pc_ functions in _Drosophila_ clock neurons.

### Analysis of TaDa data revealed different Pc functional gene profiles in clock neurons

We classified the Pc binding genes enriched in each cluster of clock neurons based on the gene ontology (GO) annotations and searched for overrepresented functional categories. Notably, the genes with nucleic acid binding and transcription regulator activities represented the largest population of the four clock neurons-enriched Pc binding genes. In the considered clusters of clock neurons, there were marked differences in the categories of enrichment for Pc binding genes. The Pc binding genes with transporter activity-related function were enriched in _R6-Gal4_–marked neurons (s-LNvs). Most of clock neurons except neurons marked by _R18H11-Gal4_–enriched Pc binding genes that have various catalytic activities (Fig 3A–D; Table S2). The above data revealed that a large number of Pc binding genes involved in different functional categories were enriched in the clock neurons.

We further compared the Pc binding genes enriched in different pathways in four clock neurons. The common Pc binding genes in

the four neuron types are transcription regulators and transcription regulatory region sequence-specific DNA bindings. However, there were marked differences in the pathways of four clock neurons-enriched Pc binding genes. For instance, Pc binding genes with signaling receptor activator, heme binding, and iron ion binding were overrepresented in _DVpdf-Gal4_, _pdf-Gal80_–marked neurons. Also, Pc binding genes with G protein-coupled amine receptor activity and peroxidase activity were overrepresented in _R18H11-Gal4_–marked neurons. Furthermore, Pc binding genes with trans-membrane signaling receptor activity and ligand-gated ion channel activity were overrepresented in _R6-Gal4_–marked neurons, whereas Pc binding genes enriched in various signaling receptor binding in _C929-Gal4_–marked neurons (Fig 3E; Table S3). The KEGG enrichment analysis also revealed that these groups of genes were enriched in various metabolism pathways such as cytochrome P450-related metabolism process, retinol metabolism, and neuroactive-ligand receptor interaction (Fig S4A; Table S3). Based on the above, cluster analysis of Pc functional targets revealed that Pc targets vary in the four clusters of clock neurons.

### Identification of Pc genomic binding sites in different clusters of clock neurons

To explore how _Pc_ is involved in circadian rhythm regulation in the considered four specific clock neuron clusters, we conducted an extensive search for circadian-related genes in previous studies (Table S4). This effort led to the compilation of a comprehensive list of 5,045 circadian-related genes from various sources, distributed across different datasets as follows: Clock Binding Gene dataset, encompassing 1,517 genes (Abruzzi et al, 2011); Clock Neuro dataset, comprising 249 genes (Nagoshi et al, 2010); CGDB (Circadian Gene Database), containing 2,662 genes (Li et al, 2017); JTK Cycle dataset, consisting of 676 genes (Li et al, 2017; Kumar et al, 2021); and the _Per_ mutant dataset, including 1,475 genes (Hughes et al, 2012). This pool of 5,045 circadian-related genes is much larger than circadian regulatory gene pool. Upsetplot was then used to analyze the correlation between the circadian-related genes and the Pc binding

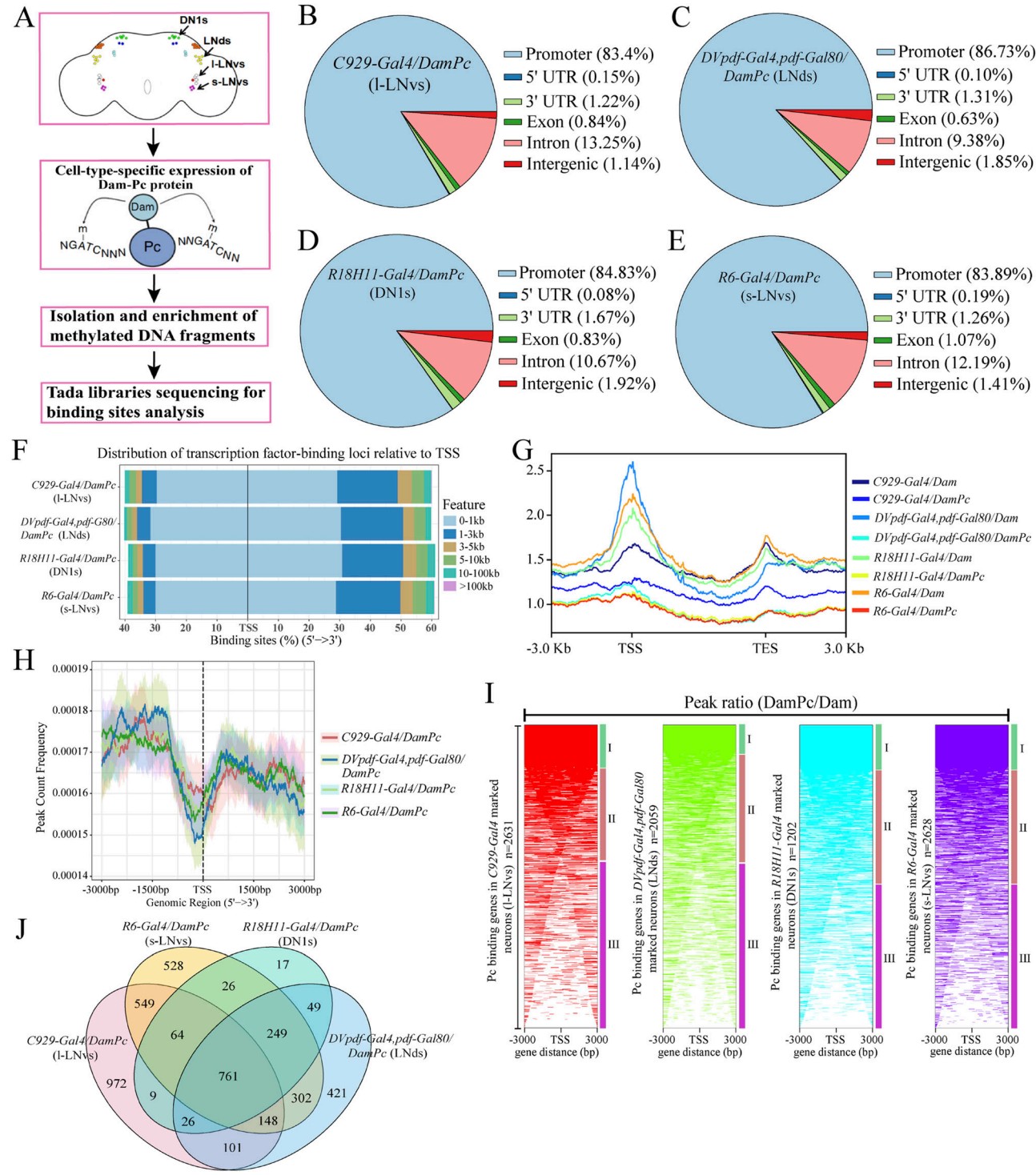

**Figure 2. Analysis of the global features of Pc binding gene profile identified by targeted DamID.**
**(A)** Experimental design of targeted DamID. **(B, C, D, E)** Distribution of annotated peaks by genomic features for all accessible regions, including distal intergenic, promoter (≤1, 1–2, and 2–3 kb), intron, exon, UTR, and other regions as shown in the legend. **(F)** Distribution of transcription factor-binding loci relative to transcriptional start sites (TSS) of differentially accessible regions for Pc binding with four clusters of clock neurons. **(G)** Density plots of average signals of Dam-Pc and Dam only at all *Drosophila melanogaster* genes, and the average normalized profile across the gene body (±3 kb) were calculated. TSS, transcription start sites; TES, transcription end sites. **(H)** Average profile of Dam-Pc/Dam peaks binding to the TSS region (±3 kb). 95% confidence interval was calculated for each cluster genes. **(I)** Heatmap profile of peaks depicting their distribution relative to the TSS considering ±3 kb regions for Pc binding genes in four clusters of clock neurons. **(J)** Venn diagram showing the overlap of genes number of Pc binding in four clusters of clock neurons (*DVpdf-Gal4*, *pdf-Gal80*, *R6-Gal4*, *C929-Gal4*, and *R18H11-Gal4*–marked neurons).

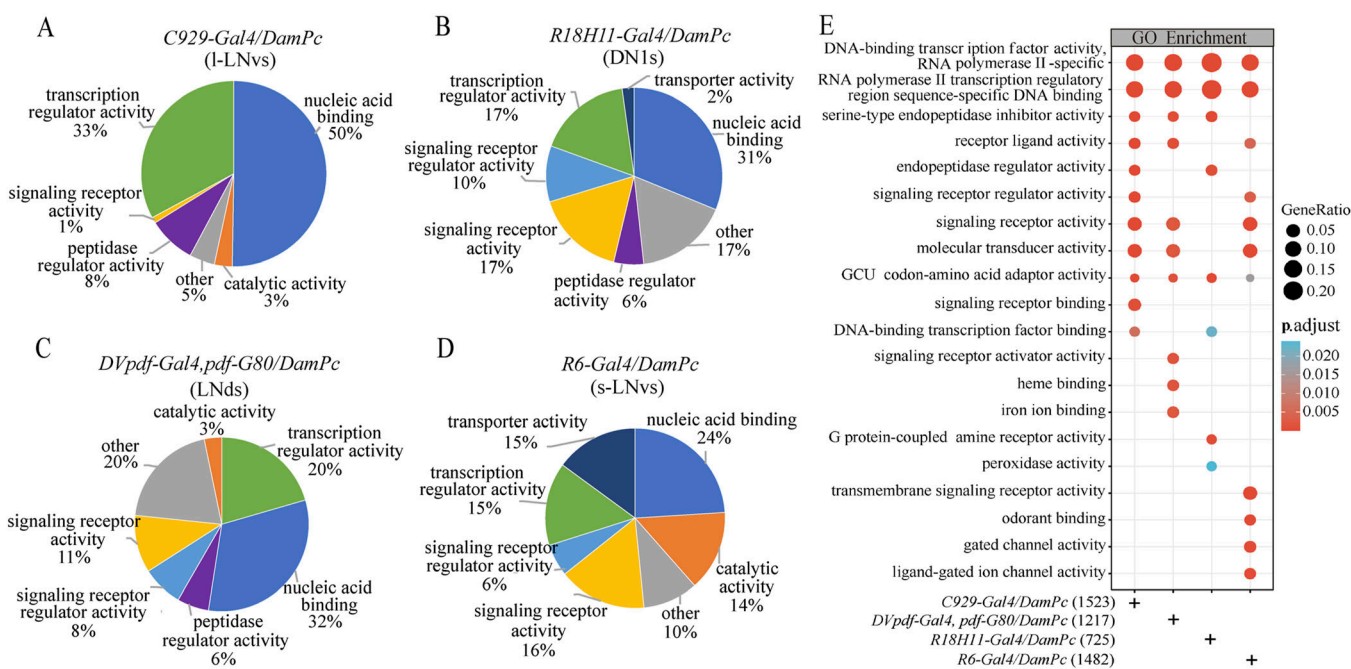

**Figure 3. Functional classifications of the Pc binding genes enriched in four clusters of clock neurons.**
**(A, B, C, D)** Functional classifications of Pc binding genes enriched in specific clock neurons. The genes were assigned to functional classes on the basis of the GO annotation. **(E)** GO enrichment analysis of Pc binding genes in four clusters of clock neurons. The results were displayed as multiple columns with each one representing an enrichment result of a clock neuron cluster. The color of each bubble represents the *P*.adjust value. The bubble size represents the gene ratio of the number of enriched genes in the GO enrichment terms to the number of annotated background genes in this term.

genes enriched in each cluster of clock neurons. It was seen that 762, 584, 701, and 302 genes had overlaps in *C929-Gal4* (l-LNvs cluster), *DVpdf-Gal4*, *pdf-Gal80* (LNds cluster), *R6-Gal4* (s-LNvs cluster), and *R18H11-Gal4*–marked (DN1s cluster) neurons respectively. There were also 412 genes with overlapping in *C929-Gal4*–marked (l-LNvs cluster) and *R6-Gal4*–marked (s-LNvs cluster) neurons, 189 genes with overlapping in all the four clusters of clock neurons, and 1,051 genes with overlapping in *C929-Gal4*–marked (l-LNvs cluster) or *R6-Gal4*–marked (s-LNvs cluster) neurons (Fig 4A; Table S4). More circadian-related Pc targets were found in *C929-Gal4* (l-LNvs cluster) or *R6-Gal4*–marked (s-LNvs cluster) neurons comparing with *DVpdf-Gal4*, *pdf-Gal80* (including LNds cluster) or *R18H11-Gal4* (including DN1s cluster)–marked neurons. In addition, we conducted separate comparisons between Pc binding and the five datasets, and an analysis of the proportion of binding genes relative to these datasets. The percentage of Pc-bound targets associated with circadian rhythms appeared to exhibit a similar pattern in each cell type (Fig S4C and D). Consistent with this, it was found that altering *Pc* levels in the *R6-Gal4*–marked (s-LNvs cluster) neurons and *C929-Gal4*–marked (l-LNvs cluster) neurons result in significant defects in the circadian rhythm (Fig 1).

To further investigate the interactions between the products of genes bound preferentially by Pc in LNvs, we generated a network of interactions based on previous literature. The network of interactions highlighted that *Pc* was involved in signaling pathways which is active in the *C929-Gal4* (l-LNvs cluster) and *R6-Gal4* (s-LNvs cluster)–marked neurons, for example, cell fate and most

developmental decisions (Fig S5). Previous studies have found that genes that are associated with similar phenotypes are in proximity to each other in protein–protein interaction (PPI) networks, suggesting that the circadian rhythm and specific neuron-associated biomarkers may also cluster within specific PPI networks. Therefore, the PPI network of 1,051 circadian-related genes based on Pc binding in the *C929-Gal4* (l-LNvs cluster) and *R6-Gal4* (s-LNvs cluster)–marked neurons (panel A red tag) was constructed in the STRING database. We used the degree and betweenness centrality metrics to identify hub genes in a PPI network. Degree captures a node's connectivity, whereas betweenness centrality gauges a node's frequency of acting as an intermediary in the network. The color from red to yellow indicates the degree of interaction from high to low (Fig 4B). The GO clustering of the 1,051 Pc binding genes indicated that the neuron projection development, cell morphogenesis, circadian rhythm and rhythmic process-related genes, etc., were significantly enriched (Fig 4C; Table S5). Further analysis revealed that the genes involved in the circadian rhythm regulation were significantly enriched (Fig 4D), for example, *period* (*per*) (Yang & Sehgal, 2001; Ceriani et al, 2002), *timeless* (*tim*) (Wang et al, 2001; Yang & Sehgal, 2001), *PAR-domain protein* 1 (*Pdp1*) (Cyran et al, 2003), *Mid1* (Ghezzi et al, 2014), *Rhodopsin 5* (R*h5*) (Szular et al, 2012), *kayak* (*kay*) (Ling et al, 2012), *nervy* (*nvy*) (Duvall & Taghert, 2013), and *B4* (Claridge-Chang et al, 2001). In addition, clustering of the 1,051 Pc binding dataset showed that they are also enriched for genes in the metabolism pathway and neuroactive ligand–receptor interaction pathway (Fig S4B; Table S5). The above data suggest that in *Drosophila*, Pc targets a set of important

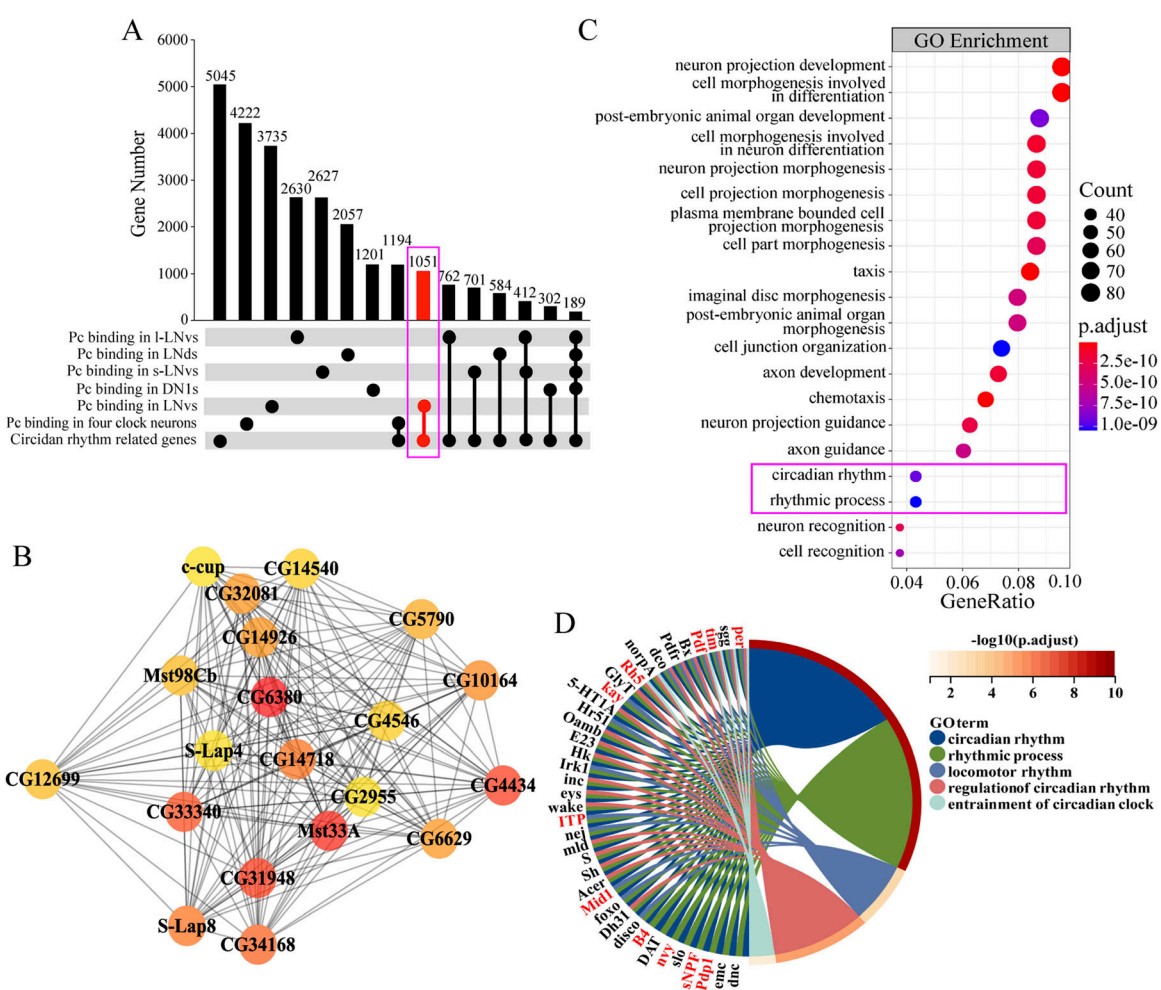

**Figure 4. Analysis of targeted DamID-seq data revealed an enrichment of clock genes in Pc target profile.**
**(A)** Gene intersection number analysis of Pc binding genes identified by targeted DamID-seq in four clusters of clock neurons. Circles indicate the genes with specific presence of the corresponding category. The circadian rhythm-related genes were based on the literature in *Drosophila melanogaster*. The corresponding gene lists can be found in the attachment file (Table S4). The circles indicated the number of genes of the correspondent category. The connected circles indicated the number of overlapping genes of correspondent category. **(B)** Protein–protein interaction analysis of top 20 hub genes of 1,051 circadian rhythm genes based on Pc binding in the s-LNvs and l-LNvs neurons. The color from red to yellow indicates the degree of interaction from high to low. **(C)** Bubble plot of GO enrichment analysis of 1,051 circadian rhythm genes based on Pc binding in the s-LNvs and l-LNvs neurons. The bubble color represents the *P*.adjust. **(D)** GO plot of top five terms involved in the rhythm regulation based on GO enrichment of Fig 4C. The color of GO term represents different terms, and the color scale bar in the upper right represents the *P*.adjust size of terms after processing, corresponding to the color of the line outside the right half of the left circle figure. The red genes of the left circle are some of the more common genes involved in rhythm which has been reported.

circadian rhythm regulatory genes in *C929-Gal4* (l-LNvs cluster) and *R6-Gal4*–marked (s-LNvs cluster) neurons.

## Pc bindings on circadian rhythm regulatory genes were predominantly found in *R6-Gal4* and *C929-Gal4*–marked neurons

To specify putative targets found by TaDa, we focused on selected candidates that were known to participate in the circadian rhythm gene network in *Drosophila*. To control the overall chromatin accessibility, signal normalization was carried out with Dam alone data, where a general pattern of each chromatin was revealed (Fig S6A–G). We presented part of these results by focusing on individual genes that were known to be involved in circadian rhythm regulation. The binding of Pc to these genes is indicated by the log₂

(Dam-Pc/Dam) value (Fig 5A–I). The results indicated that some of these genes were specifically enriched by Pc in *C929-Gal4* (l-LNvs cluster) and *R6-Gal4*–marked (s-LNvs cluster) neurons (e.g., *per*, *Rh5*, *Mid1*, *nvy*, *kay*, *B4*, and *Bx*). Previous studies also suggested that these circadian rhythm regulation-related genes are largely expressed in clock neurons such as *C929-Gal4* (l-LNvs cluster) and *R6-Gal4* (s-LNvs cluster) marked neurons (Yang & Sehgal, 2001; Ling et al, 2012; Duvall & Taghert, 2013; Ghezzi et al, 2014). It was also previously shown the core clock genes including *per*, *tim*, and *Pdp1* have a significant regulatory effect on circadian rhythm activity in *Drosophila* (Wang et al, 2001; Yang & Sehgal, 2001; Ceriani et al, 2002; Cyran et al, 2003). For peripheral circadian rhythm-related genes, *Mid1* contributes to stretch-activated, cation-selective, calcium channel activity. Knockdown of *Mid1* expression

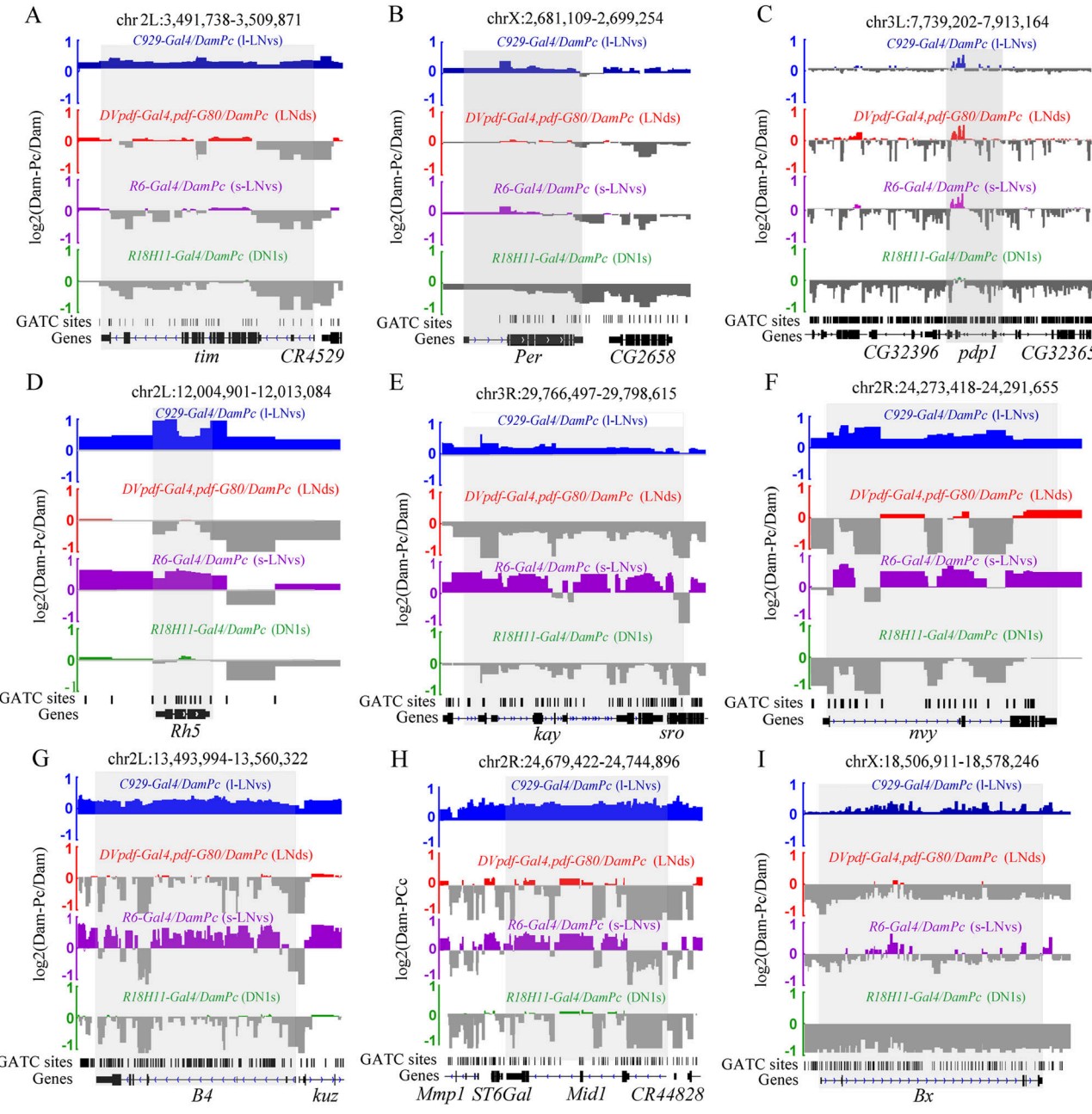

**Figure 5. DamID-seq profiling of Pc binding in four clusters of specific clock neurons.**
**(A, B, C, D, E, F, G, H, I)** Comparison profiles of representative circadian rhythm genes from targeted DamID data (From Fig 4D). Vertical bars show log₂ ratio between the Dam-Pc and Dam only signal, log₂ (Dam-Pc/Dam), the shadow area represent gene region.

also disrupts locomotor rhythm in *Drosophila* (Ghezzi et al, 2014). Furthermore, *kay* encodes a transcription factor involved in multiple biological processes including the locomotor rhythm process. Also, the role of *kay* in the transcriptional loops is to control *Drosophila* circadian behavior (Ling et al, 2012). Furthermore, *nvy* is a member of the MTG family of genes that have both nuclear and cytosolic functions encoding a transcriptional repressor involved in the circadian rhythm in *Drosophila* (Duvall & Taghert, 2013).

Remarkably, we found that a large number of neurotransmitters and neuropeptides were also enriched in four clock neurons (Table 3).

Further analysis showed that Pc binding on neuropeptides that are important in circadian regulation has specific patterns across the four clusters of clock neurons (Fig S6H–K). Instances of such binding are *pdf* (Choi et al, 2009), *short neuropeptide F* (*sNPF*) (Johard et al, 2009), *neuropeptide F* (*NPF*) (He et al, 2013), and *Ion Transport Peptide* (*ITP*) (Johard et al, 2009). Based on the previous studies, we concluded that many important molecules related to genes in neural signal transduction play an important role in circadian rhythm regulation. For instance, Pdf encodes a secreted biologically active neuropeptide that acts via a specific G-protein-coupled receptor to trigger intracellular

**Table 3. Pc-target neurotransmitters and neuropeptides in four clusters of clock neurons by targeted DamID.**

| | Gene | CG Number | Category | Presence in neurons | | | |
|---|---|---|---|---|---|---|---|
| | | | | R6-Gal4 | C929-Gal4 | Dvpdf-Gal4, pdf-Gal80 | R18H11-Gal4 |
| Glutamate system | VGLUT | CG9887 | Transporter/Vesicle transporter | + | + | + | + |
| | NMDAR1 | CG2902 | Receptor/Ion channel | − | − | − | − |
| | NMDAR2 | CG33513 | Receptor/Ion channel | − | − | − | − |
| | mGluRA | CG11144 | Receptor/GPCR | − | − | − | − |
| | GluRIA | CG8442 | Receptor/Ion channel | − | − | − | − |
| | GluRIB | CG43743 | Receptor/Ion channel | + | − | − | − |
| | GluRIIA | CG6992 | Receptor/Ion channel | + | + | + | + |
| | GluRIIB | CG7234 | Receptor/Ion channel | + | + | + | − |
| | GluRIIC | CG4226 | Receptor/Ion channel | + | + | + | + |
| | GluRIID | CG18039 | Receptor/Ion channel | − | − | − | − |
| | GluRIIE | CG31201 | Receptor/Ion channel | − | − | + | − |
| | GluClα | CG7535 | Receptor/Ion channel | − | − | − | − |
| | CG5621 | CG5621 | Receptor/Ion channel | − | − | − | − |
| | CG11155 | CG11155 | Receptor/Ion channel | − | − | − | − |
| | Ekar | CG9935 | Receptor/Ion channel | + | + | − | − |
| | CG3822 | CG3822 | Receptor/Ion channel | − | − | − | − |
| | clumsy | CG8681 | Receptor/Ion channel | − | − | − | − |
| GABA system | VGAT | CG8394 | Transporter/Vesicle transporter | + | − | − | − |
| | GAD1 | CG14994 | Synthetase | + | + | + | + |
| | Rdl | CG10537 | Receptor/Ion channel | − | − | − | − |
| | Lcch3 | CG17336 | Receptor/Ion channel | − | − | − | − |
| | GABA-B-R1 | CG15274 | Receptor/GPCR | + | − | − | − |
| | GABA-B-R2 | CG6706 | Receptor/GPCR | − | − | − | − |
| | GABA-B-R3 | CG3022 | Receptor/GPCR | + | − | − | − |
| Glycine system | GlyT | CG5549 | Transporter/Membrane transporter | + | + | − | − |
| | Grd | CG7446 | Receptor/Ion channel | − | − | − | − |
| | CG7589 | CG7589 | Receptor/Ion channel | − | − | − | − |
| | CG12344 | CG12344 | Receptor/Ion channel | − | − | − | − |
| | VAChT | CG32848 | Transporter/Vesicle transporter | + | − | − | + |
| | ChAT | CG12345 | Synthetase | + | − | + | + |
| | mAChR-A | CG4356 | Receptor/GPCR | + | + | − | − |
| | mAChR-B | CG7918 | Receptor/GPCR | + | + | − | − |
| | mAChR-C | CG12796 | Receptor/GPCR | − | − | − | − |
| | nAChR α1 | CG5610 | Receptor/Ion channel | − | − | − | − |
| | nAChR α2 | CG6844 | Receptor/Ion channel | − | − | − | − |
| | nAChR α3 | CG2302 | Receptor/Ion channel | − | − | − | − |
| | nAChR α4 | CG12414 | Receptor/Ion channel | − | − | − | − |
| | nAChR α5 | CG32975 | Receptor/Ion channel | − | − | − | − |

| | Gene | CG Number | Category | Presence in neurons | | | |
|---|---|---|---|---|---|---|---|
| | | | | R6-Gal4 | C929-Gal4 | Dvpdf-Gal4, pdf-Gal80 | R18H11-Gal4 |
| Acetylcholine system | nAChR α6 | CG4128 | Receptor/Ion channel | − | − | − | − |
| | nAChR α7 | CG32538 | Receptor/Ion channel | − | − | − | − |
| | nAChRβ1 | CG11348 | Receptor/Ion channel | − | − | − | − |
| | nAChRβ2 | CG6798 | Receptor/Ion channel | − | − | − | − |
| | nAChRβ3 | CG11822 | Receptor/Ion channel | − | − | − | − |
| Dopamine system | DAT | CG8380 | Transporter/Membrane transporter | + | − | + | − |
| | TH | CG10118 | Synthetase | − | − | − | − |
| | Dop1R1 | CG9652 | Receptor/GPCR | + | + | + | + |
| | Dop1R2 | CG18741 | Receptor/GPCR | + | + | − | − |
| | Dop2R | CG33517 | Receptor/GPCR | − | + | − | − |
| | DopEcR | CG18314 | Receptor/GPCR | − | − | − | − |
| Serotonin system | SerT | CG4545 | Transporter/Membrane transporter | − | + | − | − |
| | TrH | CG9122 | Synthetase | + | + | + | + |
| | 5-HT1A | CG16720 | Receptor/GPCR | + | − | + | − |
| | 5-HT1B | CG15113 | Receptor/GPCR | + | − | + | − |
| | 5-HT2A | CG1056 | Receptor/GPCR | − | + | − | − |
| | 5-HT2B | CG42796 | Receptor/GPCR | + | + | − | − |
| | 5-HT7 | CG12073 | Receptor/GPCR | + | + | − | − |
| Histamine system | HDC | CG3454 | Synthetase | + | + | + | + |
| | HisCl1 | CG14723 | Receptor/Ion channel | − | − | − | − |
| | Ort | CG7411 | Receptor/Ion channel | − | − | + | + |
| Octopamine/Tyramine system | TβH | CG1543 | Synthetase | − | − | − | − |
| | Oamb | CG3856 | Receptor/GPCR | + | − | + | + |
| | Oa2 | CG6919 | Receptor/GPCR | − | − | − | − |
| | Octβ2R | CG33976 | Receptor/GPCR | − | − | − | − |
| | Octβ3R | CG42244 | Receptor/GPCR | − | − | − | − |
| | Oct-TyrR | CG7485 | Receptor/GPCR | − | − | − | − |
| | TyrR | CG7431 | Receptor/GPCR | + | − | + | + |
| | TyrRII | CG16766 | Receptor/GPCR | − | + | − | + |
| Adenosine | NT5E1 | CG4827 | Synthetase | − | − | − | − |
| | NT5E2 | CG30104 | Synthetase | + | − | + | − |
| | AdoR | CG9753 | Receptor/GPCR | + | + | − | − |
| D-serine | SR | CG8129 | Synthetase | + | + | + | + |
| | CG11236 | CG11236 | D-amino-acid oxidase | − | − | − | − |
| | CG12338 | CG12338 | D-amino-acid oxidase | − | − | − | − |
| | CG3011 | Shmt | Synthetase | − | − | − | − |
| | Akh | CG1171 | Peptide | − | − | − | − |
| | AkhR | CG11325 | Receptor/GPCR | − | − | + | − |
| | AstA | CG13633 | Peptide | + | − | + | + |
| | AstA-R1 | CG2872 | Receptor/GPCR | + | + | − | − |

**Table 3. Continued**

| | Gene | CG Number | Category | Presence in neurons | | | |
|---|---|---|---|---|---|---|---|
| | | | | R6-Gal4 | C929-Gal4 | Dvpdf-Gal4, pdf-Gal80 | R18H11-Gal4 |
| | AstA-R2 | CG10001 | Receptor/GPCR | + | − | − | − |
| | AstC | CG14919 | Peptide | − | + | − | − |
| | AstC-R1 | CG7285 | Receptor/GPCR | − | − | − | − |
| | AstC-R2 | CG13702 | Receptor/GPCR | − | − | − | − |
| | Capa | CG15520 | Peptide | + | − | − | − |
| | CapaR | CG14575 | Receptor/GPCR | − | − | − | − |
| | CCAP | CG4910 | Peptide | − | − | − | − |
| | CCAP-R | CG33344 | Receptor/GPCR | + | − | − | − |
| | CCHa1 | CG14358 | Peptide | + | + | + | + |
| | CCHa1-R | CG30106 | Receptor/GPCR | + | − | + | − |
| | CCHa2 | CG14375 | Peptide | + | + | + | + |
| | CCHa2-R | CG14593 | Receptor/GPCR | − | − | − | − |
| | CNMamide | CG13936 | Peptide | − | − | − | − |
| | CNMaR | CG33696 | Receptor/GPCR | − | − | − | − |
| | Crz | CG3302 | Peptide | − | − | − | − |
| | CrzR | CG10698 | Receptor/GPCR | − | − | − | − |
| | Dh31 | CG13094 | Peptide | + | + | + | − |
| | Dh31-R | CG32843 | Receptor/GPCR | + | − | + | − |
| | Dh44 | CG8348 | Peptide | − | + | − | − |
| | Dh44-R1 | CG8422 | Receptor/GPCR | − | − | − | − |
| | Dh44-R2 | CG12370 | Receptor/GPCR | + | − | − | − |
| | Dsk | CG18090 | Peptide | − | − | − | − |
| | CCKLR-17D1 | CG42301 | Receptor/GPCR | − | − | − | − |
| | CCKLR-17D3 | CG32540 | Receptor/GPCR | − | + | − | − |
| | ETH | CG18105 | Peptide | − | − | − | − |
| | ETHR | CG5911 | Receptor/GPCR | + | − | + | + |
| | FMRFa | CG2346 | Peptide | − | − | − | − |
| | FMRFaR | CG2114 | Receptor/GPCR | − | − | − | − |
| | Lk | CG13480 | Peptide | − | − | − | − |
| | Lkr | CG10626 | Receptor/GPCR | − | + | − | − |
| | Mip | CG6456 | Peptide | − | − | − | − |
| | SPR | CG16752 | Receptor/GPCR | + | + | − | − |
| | Ms | CG6440 | Peptide | + | − | + | + |
| | MsR1 | CG8985 | Receptor/GPCR | − | − | − | − |
| | MsR2 | CG43745 | Receptor/GPCR | − | + | − | − |
| | NPF | CG10342 | Peptide | − | − | + | − |
| | NPFR | CG1147 | Receptor/GPCR | − | − | − | − |
| | Pdf | CG6496 | Peptide | + | + | − | − |
| | Pdfr | CG13758 | Receptor/GPCR | − | + | − | − |
| | Hug | CG6371 | Peptide | + | − | − | − |

| | Gene | CG Number | Category | Presence in neurons | | | |
|---|---|---|---|---|---|---|---|
| | | | | R6-Gal4 | C929-Gal4 | Dvpdf-Gal4, pdf-Gal80 | R18H11-Gal4 |
| | PK1-R | CG9918 | Receptor/GPCR | + | + | + | + |
| | PK2-R1 | CG8784 | Receptor/GPCR | + | + | + | + |
| | PK2-R2 | CG8795 | Receptor/GPCR | + | + | + | + |
| | Proc | CG7105 | Peptide | − | + | + | − |
| | Proc-R | CG6986 | Receptor/GPCR | − | + | − | − |
| | RYa | CG40733 | Peptide | − | − | − | − |
| | RYa-R | CG5811 | Receptor/GPCR | + | + | − | − |
| | SIFa | CG33527 | Peptide | − | − | − | − |
| | SIFaR | CG10823 | Receptor/GPCR | − | − | − | − |
| | sNPF | CG13968 | Peptide | + | + | − | − |
| | sNPF-R | CG7395 | Receptor/GPCR | − | − | − | − |
| | Tk | CG14734 | Peptide | + | + | + | + |
| | TkR86C | CG6515 | Receptor/GPCR | + | + | + | + |
| | TkR99D | CG7887 | Receptor/GPCR | + | + | − | − |
| | Trissin | CG14871 | Peptide | − | − | − | − |
| | TrissinR | CG34381 | Receptor/GPCR | + | + | + | − |
| | Burs | CG13419 | Peptide | − | − | − | − |
| | rk | CG8930 | Receptor/GPCR | − | + | − | − |
| | Ptth | CG13687 | Peptide | + | + | + | − |
| | Nplp4 | CG15361 | Peptide | − | − | − | − |
| | Dup99B | CG33495 | Peptide | + | + | − | − |
| | Nplp3 | CG13061 | Peptide | − | − | − | − |
| | Nplp1 | CG3441 | Peptide | + | + | + | + |
| | CG8216 | CG8216 | Peptide | − | − | − | − |
| | natalisin | CG34388 | Peptide | − | + | + | − |
| | amn | CG11937 | Peptide | − | + | − | − |
| | Eh | CG5400 | Peptide | − | − | + | − |
| | ITP | CG13586 | Peptide | − | + | − | − |
| | Orcokinin | CG13565 | Peptide | − | − | − | − |
| | Pburs | CG15284 | Peptide | + | + | − | − |
| | Nplp2 | CG11051 | Peptide | − | − | − | − |
| | CG13229 | CG13229 | GPCR/Orphan | + | − | − | − |
| | CG13575 | CG13575 | GPCR/Orphan | − | − | − | − |
| | CG13995 | CG13995 | GPCR/Orphan | − | + | − | − |
| | CG30340 | CG30340 | GPCR/Orphan | − | − | − | − |
| | CG32547 | CG32547 | GPCR/Orphan | − | − | − | − |
| | CG33639 | CG33639 | GPCR/Orphan | − | + | − | − |
| | CG34411 | CG34411 | GPCR/Orphan | − | − | − | − |
| | CG13579 | CG13579 | GPCR/Orphan | + | + | − | − |
| | CG15744 | CG15744 | GPCR/Orphan | − | − | − | − |
| | mthl6 | CG16992 | GPCR/Orphan | + | − | + | − |

**Table 3. Continued**

| | Gene | CG Number | Category | Presence in neurons | | | |
|---|---|---|---|---|---|---|---|
| | | | | R6-Gal4 | C929-Gal4 | Dvpdf-Gal4, pdf-Gal80 | R18H11-Gal4 |
| | mthl10 | CG17061 | GPCR/Orphan | − | − | − | − |
| | mthl13 | CG30018 | GPCR/Orphan | − | − | − | − |
| | mthl11 | CG31147 | GPCR/Orphan | − | − | − | − |
| | mthl2 | CG17795 | GPCR/Orphan | − | − | − | − |
| | CG31760 | CG31760 | GPCR/Orphan | + | + | − | − |
| | CG32447 | CG32447 | GPCR/Orphan | − | − | − | − |
| | mthl8 | CG32475 | GPCR/Orphan | + | − | − | − |
| | CG43795 | CG43795 | GPCR/Orphan | + | + | + | − |
| | CG44153 | CG44153 | GPCR/Orphan | + | + | + | − |
| | mth | CG6936 | GPCR/Orphan | − | − | − | − |
| | mthl7 | CG7476 | GPCR/Orphan | − | − | − | − |
| | CG11318 | CG11318 | GPCR/Orphan | − | − | − | − |
| | CG7497 | CG7497 | GPCR/Orphan | − | − | − | − |
| | CG12290 | CG12290 | GPCR/Orphan | + | − | − | − |
| | mthl1 | CG4521 | GPCR/Orphan | − | − | − | − |
| | mthl3 | CG6530 | GPCR/Orphan | − | − | + | − |
| | mthl4 | CG6536 | GPCR/Orphan | − | − | − | − |
| | mthl5 | CG6965 | GPCR/Orphan | − | − | − | − |
| | CG15556 | CG15556 | GPCR/Orphan | − | − | − | − |
| | CG15614 | CG15614 | GPCR/Orphan | + | − | − | − |
| | mthl9 | CG17084 | GPCR/Orphan | − | − | − | − |
| | CG18208 | CG18208 | GPCR/Orphan | − | − | − | − |
| | mtt | CG30361 | GPCR/Orphan | − | − | − | − |
| | pog | CG31660 | GPCR/Orphan | − | − | − | − |
| | mthl15 | CG31720 | GPCR/Orphan | − | − | − | − |
| | mthl14 | CG32476 | GPCR/Orphan | − | − | − | − |
| | mthl12 | CG32853 | GPCR/Orphan | − | − | − | − |
| | CG4313 | CG4313 | GPCR/Orphan | − | − | − | − |
| | hec | CG4395 | GPCR/Orphan | − | − | − | − |
| | Lgr1 | CG7665 | GPCR/Orphan | − | − | + | − |
| | Lgr3 | CG31096 | GPCR/Orphan | − | − | − | − |
| | moody | CG4322 | GPCR/Orphan | − | − | − | − |
| | stan | CG11895 | GPCR/Orphan | − | − | − | − |
| | Tre1 | CG3171 | GPCR/Orphan | − | − | − | − |

signaling. It has a prominent role in the neural circuit controlling circadian rest–activity rhythms in *D. melanogaster* (Peng et al, 2003; Miyasako et al, 2007). Also, sNPF encodes a protein that binds to the product of sNPF-R and activates ERK-Dilps signaling or the PKA-CREB pathway. Its roles include regulation of carbohydrate metabolism, locomotion, and the circadian rhythm (Johard et al, 2009; Chen et al, 2013; Shang et al, 2013). These findings reveal that Pc-binding gene regulates the circadian activity rhythms, suggesting that the

phenotype of *PcRNAi* was the result of the collective effects of its target genes.

Moreover, we validated these results by PCR on *Per* and *Tim* that were known to be involved in circadian rhythm regulation after TaDa experiments in different Gal4s. The Primers used on the gene locus were shown (Fig 6A). We used *iab-7* which was a Pc target identified before (Mihaly et al, 1997) as control for the binding. The PGRP-LE served as a negative control, a non-PcG target (Du et al, 2016). The

results indicated that in *C929-Gal4*, *pdf-Gal4*– and *R6-Gal4*–marked neurons, Pc had binding comparable with iab-7 on both *Per* and *Tim* gene locus (Fig 6B–D and G–I), whereas Pc did not bind to PGRP-LE (Fig 6B–D and G–I). Whereas in *DVpdf-Gal4*, *pdf-Gal80*–marked neurons, Pc had binding comparable with *iab-7* on *Tim* but not on *Per* gene locus (Fig 6F and K). In *R18H11-Gal4*–marked neurons, Pc had much less binding on both *Per* and *Tim* gene loci (Fig 6E and J). We found that the knocking down of *Pc* resulted in *Per and Tim* expression changes. At *tim-Gal4*–marked neurons, the *Pc* knocking down resulted in increased *Per* and *Tim* expression. Whereas in *C929-Gal4*, *pdf-Gal4*–, *R18H11-Gal4*–, and *Dvpdf-Gal*, *pdf-Gal80*–marked neurons, the *Per* and *Tim* expression were down-regulated by *Pc* knocking down (Fig 6L–U, see the Discussion section).

To gain insight into PC binding differences between clock neurons and non-clock neurons, we checked Pc bindings and H3K27me3 at specific gene loci by conducting TaDa assays and ChIP assays followed by PCR, respectively. The experiments were done at *zfh2*, *vnd* gene locus which were positively bound by Pc and *mthl10*, *prosbeta2* gene locus which were not bound by Pc based on TaDa-seq assay in this study. The results indicated that Pc positively bound on *zfh2*, *vnd* gene loci and did not bind on *mthl10*, *prosbeta2* gene loci in clock neuron clusters (Fig 7A–T), whereas Pc positively bound on all four loci in non-clock neurons (Fig 7A–T). Furthermore, to verify the histone modification on these sites, the ChIP experiments done in these two populations of neurons isolated by *UAS-unc84GFP* showed that the H3K27me3 modifications were prominent on all four gene loci in both clock and non-clock neurons (Fig 7A–T). The existence of H3K27me3 modifications on PC negative locus may be deposited at earlier developmental stages. The above results collectively suggested that the Pc binding detected by TaDa assay were functional.

## Discussion

Our study identified the function of *Pc* in the circadian rhythm gene network. In light of the presented TaDa data and previous works, we presented an updated gene network and proposed that *Pc* may play a prominent role in mediating circadian rhythms in the specific clusters of clock neurons. It was shown that in *Drosophila*, *Pc* is involved in sleep regulation (Zhao et al, 2021a, 2021b), neurogenesis (Bello et al, 2007), and neuron remodeling (Wang et al, 2006). However, its link to circadian rhythms in the clock neurons was not previously understood. By applying TaDa to profile Pc binding genes in specific clock neuron clusters of the *Drosophila* adult brain, we identified Pc binding genes in four specific clock neurons including s-LNvs, l-LNvs, LNds, and DN1s. The four sets of Pc binding gene profiles were more different than expected. The genes related to circadian regulation were also found in the Pc binding genes identified by comparing with the circadian gene identified in related literature. Our findings further indicated the genetic mechanism of *Pc* in regulating circadian rhythm through binding to clock target genes and clock genes loci such as *per*, *tim*, and *Pdp1*. This study is the first attempt to analyze the Pc binding gene profile of the clock neurons.

TaDa is designed to express Dam-fusion proteins at very low levels in a spatially and temporally restricted manner. Conditional activation of specific Gal4s only at the adult stage by including *Gal80^{ts}* made spatially and temporally expression possible. This enables the identification of protein–DNA interactions in a cell-specific fashion in vivo without the need for cell purification (Southall et al, 2013; Marshall & Brand., 2017; Cheetham et al, 2018; Wade et al, 2021). Currently, ChIP-seq is the most commonly used method to identify binding targets (Kudron et al, 2018; Celniker et al, 2009). Recent studies presented here indicated that TaDa is comparable with ChIP-seq in identifying target genes (Katsanos & Barkoulas, 2022). In addition, TaDa offers the following two key advantages. The first advantage is that TaDa enables the identification of DNA- or chromatin-binding in a specific tissue of interest without cell isolation, fixation or affinity purification. The second advantage is TaDa's high sensitivity, hence requiring substantially less amount of material than that of the ChIP-seq. TaDa can be applied to the study of chromatin-associated factors (van Steensel et al, 2001; Tolhuis et al, 2006; Filion et al, 2010), including transcription factor (TF) binding and assessment of chromatin states.

We performed multiple lines of evaluation to assess the quality of TaDa-seq. Using TaDa-seq, we first showed that the defined Pc targets exhibit significant abundance in the corresponding gene loci (Fig 5). We also used TaDa-seq for analyzing the defined interactions to reveal general bioinformatics features of Pc binding based on the normalized aligned read count plots and principal component analysis (PCA) (Fig S3). Finally, we showed that the functional enrichment analysis of Pc binding genes (e.g., *per*, *tim*) is in agreement with previous phenotypic studies of these two circadian rhythm-related genes (Yang & Sehgal, 2001). These confirm the reliability of the dataset of Pc–target interactions, which lays the foundation for further understanding of *Pc*.

The Gal4s used were the best we can find, although in some cases, non-clock neurons were targeted. Thus, we cannot completely exclude the possibility that we found some Pc targets in non-clock neurons. *C929-Gal4*–marked neurons labeled l-LNvs as the only clock neurons, but it has expression in other non-clock neurons (Taghert et al, 2001). However, study showed that these non-clock neurons were not involved in manipulating circadian rhythms (Grima et al, 2004). *R6-Gal4*–marked neurons have the strongest signal in s-LNvs, although several other neurons were weakly labeled (Helfrich-Förster et al, 2007). *R6-Gal4* also labeled all tracheal cells, the influence of which was minimized because we only collected samples from fly heads in our study. *R18H11-Gal4* marked DN1s neurons specifically (Guo et al, 2016). *DVpdf-Gal4*, *pdf-Gal80* marked LNds neurons specifically (Schubert et al, 2018). This may be the reason for the result showed in Fig 6L–U.

Our data represent a genome-wide identification of Pc binding genes. It however remains to be determined whether Pc binding genes within cell types at the single-cell level are heterogeneous. With recent advances in single-cell technology, future experiments may be able to better characterize variability in Pc binding genes within cell populations in circadian regulation.

The sampling time point for TaDa in this study is at ZT12. Previous studies showed that the expression level of clock genes oscillates (reviewed in Patke et al [2020]). At ZT12, most of the mRNA levels of the clock genes are at their extremes, either at the peak or the trough (Patke et al, 2020). Pc function can be either activation or inhibition (Morey & Helin, 2010; Lv et al, 2017). To the best of our

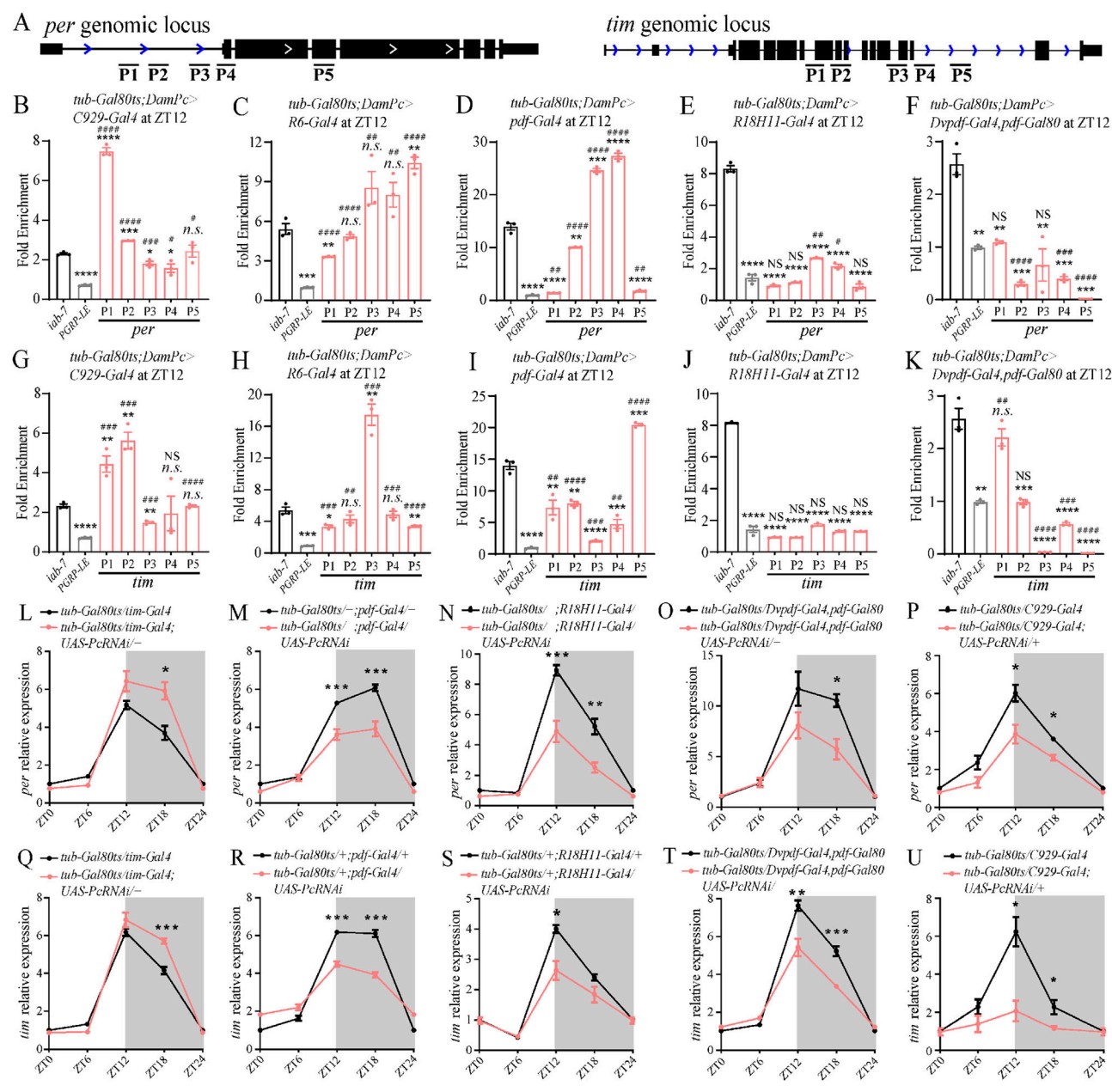

**Figure 6. Verification of the regulation of Pc target genes.**
**(A, B, C, D, E, F, G, H, I, J, K)** Dam-ID experiment to identify the binding of Pc on *tim* and *per* genomic locus. **(A)** Primers used were shown in (A). **(B, C, D, E, F, G, H, I, J, K)** The fold enrichment of different sites was shown in (B, C, D, E, F, G, H, I, J, K), the iab-7 and PGRP-LE served as positive and negative controls, respectively. **(L, M, N, O, P, Q, R, S, T, U)** *tim* and *per* are regulated by Pc in some clock neurons. Levels of Per and Tim are altered in flies with *Pc* down-regulation. Real-time quantitative RT–PCR analysis of total RNA prepared from adult brains of the indicated genotypes at the indicated time points. The relative expression levels were normalized to *Rp49* expression levels. Data information: data represent mean ± SEM. **(B, C, D, E, F, G, H, I, J, K, L, M, N, O, P, Q, R, S, T, U)** Statistical differences were measured using unpaired *t* test. n.s./NS indicates no significant difference. */# indicates $P < 0.05$, **/## indicates $P < 0.01$, ***/### indicates $P < 0.001$. */n.s. compared with the *iab-7* control, #/NS compared with the *PGRP-LE* control.

understanding, PcG complex can stabilize gene expression during the stress context (Zhao et al, 2021a, 2021b). So, we speculate that the *Pc* function may be prominent at least for these known clock genes at ZT12 when the expression levels are at their extremes. In addition, we checked the Pc binding on *Per* locus at four time points across 24 h. The results showed that the Pc binding was not statistically oscillated (Fig S7A–F). However, Pc binding *Per* P1 was

statistically oscillated in *C929-Gal4*–marked neurons (Fig S7G). Although we do not know the status on other gene locus, at least ZT12 represented most of the situations in the case of *Per*.

The relationship between expression of the Pc downstream targets and Pc bound targets were complex. Pc has direct downstream targets (bound by Pc) and indirect downstream targets (not bound by Pc). As shown in Fig 6L–U, we checked the expression level

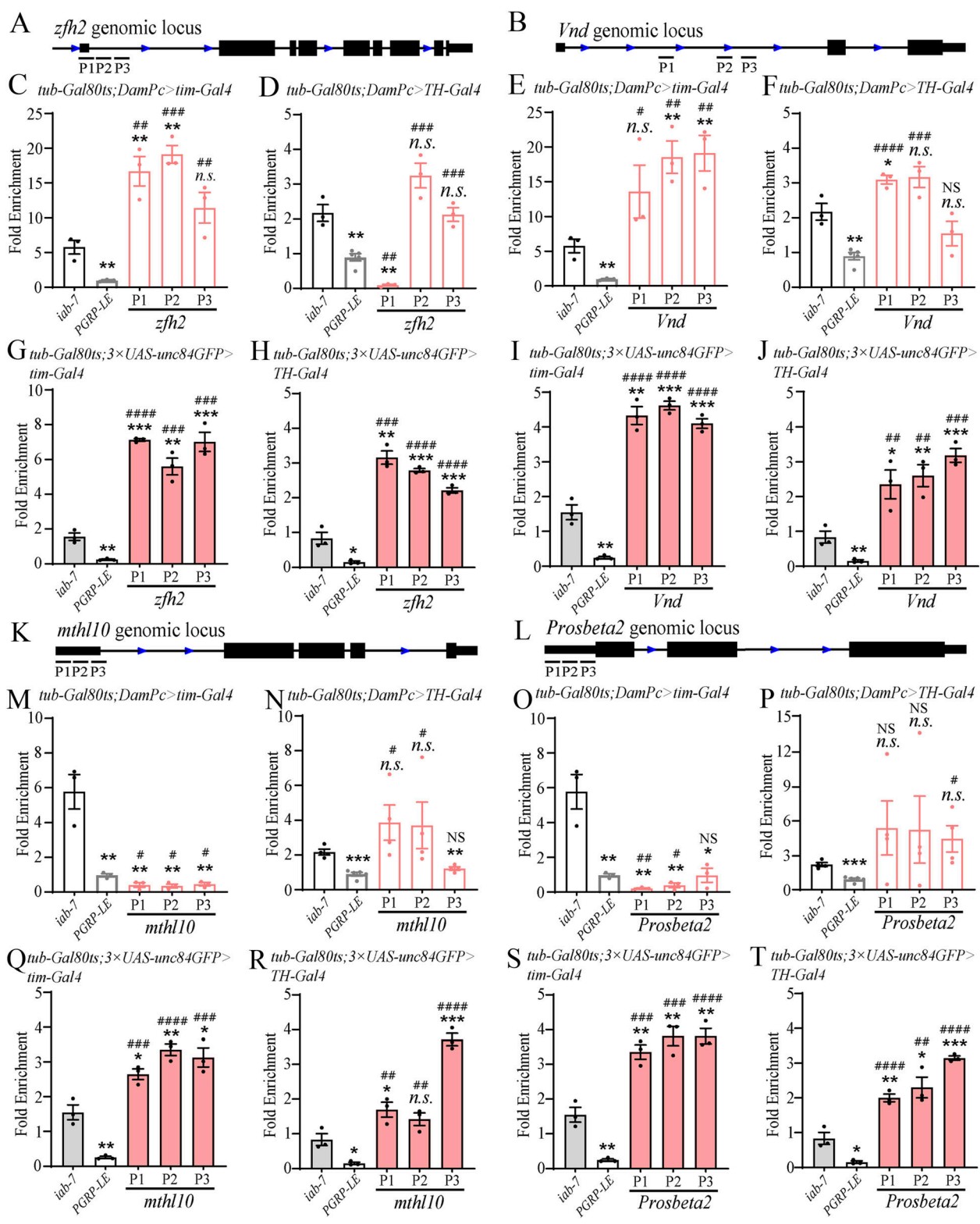

**Figure 7. Targeted DamID (TaDa) and ChIP assay followed by PCR to check Pc binding sites and the histone modification on some gene locus.**
TaDa and ChIP assay to identify the PC binding and H3K27me3 histone modification on *zfh2*, *Vnd*, *mthl10*, and *Probeta2* genomic locus. **(A, B, K, L)** Primers used were shown in (A, B, K, L). **(C, E, F, M, N, O, P)** The fold enrichment of different sites in TaDa assay, the *iab-7* and *PGRP-LE* served as positive and negative controls, respectively. **(G, H, I, J, Q, R, S, T)** The fold enrichment of different sites in ChIP experiments detecting H3K27me3, the *iab-7* and *PGRP-LE* served as positive and negative controls, respectively. Data information: data represent mean ± SEM. **(C, D, E, F, G, H, I, J, M, N, O, P, Q, R, S, T)** Statistical differences were measured using unpaired *t* test. n.s./NS

of clock genes. *Pc* can function as both repressor or activator (Morey & Helin, 2010; Lv et al, 2017). Moreover, as discussed above, in some cases, non-clock neurons were targeted in this study. So, when *Pc* is knockdown, its direct downstream target genes can be either activated or repressed under different conditions in this study. That is why we see either up-regulation or down-regulation of *Pc* targets. Pc could also have indirect target, whose expression could be regulated by direct downstream targets of Pc. This could explain what we see in *tim* and *per* in *R18H11-Gal4*–marked neurons.

Clustering of the Pc targets of the considered clusters of clock neurons reflected diversified functions of these neurons. Pc binding genes highly enriched in the four specific clock neurons are identified as having transcriptional regulator or DNA-binding activity (Fig 3). This further highlights the importance of the transcription regulation and transcriptional feedback loop in the core regulatory mechanism. The differential enrichment of Pc-binding genes in DN1s, LNvs, and LNds neurons reflects different contributions of these types of neuron clusters to the circadian circuit. Previous studies showed that the clock neurons can be classified into morning and evening neurons according to their corresponding activities (Grima et al, 2004; Stoleru et al, 2004). DN1s neurons are known to be involved in the light input and behavioral output pathway (Murad et al, 2007; Stoleru et al, 2007). In addition, glutamate-releasing DN1s can directly inhibit key pacemaker neurons (Guo et al, 2016). We argue that the newly identified Pc binding genes specifically enriched in each of the four clusters of clock neurons help define their additional functions and underlying mechanisms.

We analyzed the interactions between the circadian rhythm-related *Pc* target genes in *C929-Gal4* (including l-LNvs cluster) and R6-Gal4 (including s-LNvs cluster)–marked neurons, which collectively represents neurons including LNvs (Fig S4). The results highlighted the enrichment of genes related to signaling pathways such as cell fate and most of the developmental decisions possibly in LNvs. Among the 1,051 Pc binding circadian rhythm-related genes in this cluster of neurons (Fig 4), the presence of key clock regulatory genes such as *per* (Yang & Sehgal, 2001; Ceriani et al, 2002), *tim* (Wang et al, 2001; Yang & Sehgal, 2001), *Pdp1* (Cyran et al, 2003) is also remarkable. These findings are consistent with our findings about *Pc* function in clock neurons. Our identification of the interaction network of Pc binding genes involved in circadian systems provides a basis for studying the role of epigenetic factors in circadian regulation.

In this study, for the first time, we report that *Pc* is involved in circadian regulation through its targets in clock neurons. We found that *Pc* loss in *tim*-expressing neurons affected circadian rhythms. Down-regulating *Pc* in LNvs also has a significant effect on circadian rhythm. In contrast, down-regulating *Pc* in LNds or DN1s results in no significant circadian rhythm phenotypes. We propose that down-regulating *Pc* in the clock neurons results in varying degrees of rhythmic phenotype, possibly because of the different target genes of *Pc* in the clock neurons. We also found that Pc targets in the neuron cluster including LNvs enriched a large number of

circadian rhythm and rhythmic process-related genes, such as core clock genes, neurotransmitters, and neuropeptides, which can mediate the circadian rhythm phenotype caused by *Pc* down-regulation. Pc target genes in cluster including DN1s and LNds may be less relevant to circadian rhythms. However, as the circadian phenotype of *Pc* in neuron cluster including LNvs did not fully represent its phenotype in *tim*-expressing neurons, *Pc* function in other clock neurons may contribute to circadian regulation. The overexpression of *Pc* did not cause an obvious phenotype in contrast to *Pc* RNAis. This may be because that stringent regulation exists in the Pc binding to chromatin. In normal conditions, the Pc protein is already saturated in the system.

# Materials and Methods

## Fly stocks

The following flies were used in this study: $w^{1118}$ (NO.5905), *UAS-dicer* (NO. 24650), and *pdf-Gal4* (NO. 41286) (He et al, 2020) were obtained from Bloomington Stock Center; *UAS-PcRNAi* (NO. THU1306) (Ni et al, 2011) was obtained from TsingHua Fly Center; *TH-Gal4* (Abruzzi et al, 2017), *R18H11-Gal4* (NO.48832) (Dionne et al, 2018), *DVpdf-Gal4*, *pdf-Gal80* (Nagoshi et al, 2010), *R6-Gal4* (Helfrich-Förster et al, 2007), *C929-Gal4* (Shafer & Taghert, 2009), and *iso; tim-Gal4* (Xia et al, 2020) were gifts from Dr. Yi Rao's laboratory (Peking University, China). *UAS-FLAG-Pc* (Du et al, 2016) was a gift from Dr. Alan Jian Zhu's laboratory (Peking University, China). 3 × *UAS-unc84-GFP* (Agrawal et al, 2019) was a gift from Dr. Kent's laboratory (York University, Canada). All fly crosses were maintained at 25°C or 18°C and 60% relative humidity on a standard cornmeal–yeast–agar medium in a 12-h light/12-h dark cycle.

In this study, down-regulation of *Polycomb* (*Pc*) levels by *Pc* RNAi in *Drosophila* clock neurons (*tim-Gal4* marked most of the clock neurons) resulted in abnormal circadian rhythms. To further elucidate which clock neurons are required for the *Pc* gene to regulate the circadian rhythm, we selected several Gal4s specifically expresses in key clusters of clock neurons to drive the *Pc* RNAi. These drivers include *C929-Gal4* (marked clusters including l-LNvs), *R6-Gal4* (marked clusters including s-LNvs), *R18H11-Gal4* (marked clusters including DN1), and *DVpdf-Gal4*, *pdf-Gal80* (marked clusters including LNds). Based on the purpose of our experiments, we checked the phenotypes of flies with altered *Pc* levels in LD/DD conditions.

## *Drosophila* activity monitor-based method for circadian rhythm measurement

Locomotor activity experiments were performed using the *Drosophila* Activity Monitoring (DAM) system (Trikinetics) as previously described (Levine et al, 2002). Individual 2–3 d male flies were loaded into monitoring tubes (length, 65 mm; inner diameter,

---

indicates no significant difference. *$^{/\#}$ indicates $P < 0.05$, **$^{/\#\#}$ indicates $P < 0.01$, ***$^{/\#\#\#}$ indicates $P < 0.001$. */n.s. compared with the *iab-7* control, $^{\#}$/NS compared with the *PGRP-LE* control.

5 mm), which contained standard fly food. Tubes were plugged with cotton wool and placed in DAM to record locomotor activity during the experiment. Environmental conditions were programmed in light and temperature-controlled incubators. Light–dark conditions were always programmed as 12:12 h rectangular cycles. Because female flies can lay eggs, it is not convenient to do long-term behavior experiments, so we chose male flies to start our study.

For behavioral experiments at 25°C conditions, the flies were entrained to 12 h light:12 h dark cycles (3,600 lux of light) for 3 d and released to constant darkness for 7 d to measure periodicity. For behavioral experiments with *tub-Gal80^ts*, to prevent *Pc* down-regulation during development, flies were raised at 18°C in LD cycle. The newly eclosed male flies were transferred to 29°C to allow Gal4 activity, locomotor activity behavior was recorded 3 d at 29°C in LD condition and released to constant darkness for 7 d.

All circadian rhythm data including period, power, and rhythmicity were analyzed using FaasX software (https://trikinetics.com/, Developed by M. Boudinot) and MATLAB R2008a (https://www.mathworks.com/; MathWorks). For the circadian analysis, activity counts for each fly were binned every 30 min. Power represents the relative strength of the rhythm during DD. Flies with a power value greater than or equal to 10, along with a "width" value of 1.5 or more (indicating the number of peaks above the periodogram 95% confidence line in 30-min increments), were considered rhythmic. Similarly, for DD rhythmicity, rhythmic flies were defined as those with a minimum period peak power of 10 or above. Period calculations took into account all flies with power-significance values greater than or equal to 10. Periods were calculated for each individual fly using X2 periodogram analysis and then pooled to obtain a group average for each genotype.

## TaDa sample preparation and data analysis

### TaDa sample preparation

The experimental design of TaDa assay in this experiment is to identify Pc binding in the four clusters of neurons, which was achieved by expressing Dam-Pc, specifically in certain neuron clusters. The general background bindings were eliminated by including of Dam lines (e.g., *C929-Gal4/tub-Gal80^ts*; *UAS-Dam/+* et al.). Tada plasmids were obtained from Brand's laboratory (Marshall et al, 2016). *UAS-DamPc* was generated in our previous study; we cloned the *Pc* cDNA into the pUASTattB-LT3-Dam vector, and generated the *UAS-LT3-DamPc Drosophila* located on chromosome 3 by microinjection (Zhao et al, 2021b). The flies were reared at 21°C until eclosion. The newly eclosed flies were heat shocked for 72 h at 29°C to deactivate the Gal80. Fly head tissue was collected at ZT12. Genomic DNA was extracted from about 100 fly heads. Then the Tada experiments were performed as described by Marshall et al. with the following modifications (Marshall et al, 2016). The resulting DNA of the first round of PCR amplification was used as template for quantitative real-time PCR. All primers used are listed in Table S6. A fold enrichment was calculated as follows: Fold enrichment = $2^{(\text{CT GenoDam Primer } - \text{CT GenoDam PGRP-L}) - (\text{CT GenoDamPc Primer} - \text{CT GenoDamPc PGRP-L})}$.

The final genotypes for TaDa samples:
*C929-Gal4/tub-Gal80^ts*; *UAS-DamPc/+*;
*C929-Gal4/tub-Gal80^ts*; *UAS-Dam/+*;
*DVpdf-Gal80, pdf-Gal4/tub-Gal80^ts*; *UAS-DamPc/+*;
*DVpdf-Gal80, pdf-Gal4/tub-Gal80^ts*; *UAS-Dam/+*;
*R6-Gal4/tub-Gal80^ts*; *UAS-DamPc/+*;
*R6-Gal4/tub-Gal80^ts*; *UAS-Dam/+*;
*tub-Gal80^ts/+*; *UAS-DamPc/R18H11-Gal4*;
*tub-Gal80^ts/+*; *UAS-Dam/R18H11-Gal4*.

### Mechanism of Tada assays

TaDa is based on DamID, a technology to detect genome-wide binding sites of proteins in vivo. By creating a fusion of Dam::Pc, Dam will now add methyl groups to GATC sites near where Pc binds, allowing for sequencing to detect these methylated sites and therefore identify genomic Pc binding sites. By coupling it with the GAL4 system in *Drosophila*, it can achieve both temporal and spatial resolution. TaDa ensures that Dam-fusion protein is expressed at a very low level by using Gal80^ts system, avoiding potential toxicity and overexpression artefacts.

### TaDa data analysis

The damID-seq_pipeline was used to analyze Tada data (Marshall & Brand, 2015; Marshall et al, 2016). Transcriptome sequencing data were compared with *Drosophila* genome annotation release 6.37 using BWA software, genome sequence was divided into 395,891 intervals of different lengths according to methylated GATC sites. We calculated the ratio of experimental group coverage ratio (DamPc) to control group coverage ratio (Dam alone) in each interval. To make sure the result is accurate, we also used MACS2 software to predict the enrichment of transcripts on the genome, and then we intersected DamID-seq_pipeline and MACS2 results. Finally, we obtained a number of Pc binding target genes in the 4 clock neurons (Table S1).

## Tada data processing and peak calling

Raw data of fastq format were first processed by removing reads containing adapter, reads containing poly-N and low-quality reads to obtain clean data. The clean data were aligned to the *D. melanogaster* reference genome release dm6 using Bowtie2 (version 2.3.4.3) (Langmead & Salzberg, 2012). The resulting sam files were converted into bam files, sorted and indexed using samtools (version 1.9) (Li et al, 2009).

A previous study reported the software pipeline damID-seq pipeline script (http://owenjm.github.io/damidseq_pipeline/, Marshall & Brand, 2015), which can use bam files as input, and generate the outputs of final log$_2$ ratio files in bedGraph format. The pipeline bins the mapped reads into GATC-fragments according to GATC-sites indicated by a gff file for the dme_r6 fly genome (already provided by the authors of the pipeline on the website above) and normalizes reads against the Dam controls, in our case, the *C929-Gal4/+; DamPc/+*, *DVpdf-Gal4, pdf-Gal80/+; DamPc/+, R6-Gal4/+; DamPc/+*, and *R18H11-Gal4/DamPc* samples. The bedgraph files were visualized using the tool Integrative Genome Viewer (IGV, version 2.5.2) to extract representative tracks. MACS2 (v2.1.2) (Zhang et al, 2008) was used to call broad peaks for every dam-fusion/dam pair bedgraph files generated by the damID-seq pipeline. The R

package ChIPseeker (Yu et al, 2015) was then used to retrieve the nearest genes around the peaks in R studio (version 1.0.44) with annotation for the release of six of the *Drosophila* melanogaster genome by identifying the closest TSS in either direction of a gene along the linear genome.

### GO and KEGG enrichment analysis

GO enrichment and KEGG enrichment of Pc binding genes in four specific clock neurons were done using clusterprofiler (Wu et al, 2021), with a Benjamin–Hochberg-adjusted *P*-value of 0.05 was used as a cutoff. To further compare the similarities and differences between the four cluster genes bound by Pc, we performed functional enrichment analysis using compareCluster. At the same time, we used the simplify method to simplify output from compareCluster by removing redundancy of enriched GO terms. The bar plot and bubble plots of significant (*P* < 0.05) enrichment output result from GO and KEGG were plotted by ClusterProfiler.

### Other data visualization

The sample correlations and PCA were assessed with deepTools (Ramírez et al, 2014) (multiBigWigSummary, plotCorrelation, and plotPCA commands with bam file). And then the bam files were converted to the bigwig file format by deeptools (bamCoverage command) to obtain matrix, which was used to generate heatmaps (plotHeatmap command) and metaplots (plotProfile command) of each dam-fusion and dam only reads coverages for regions up to 3 kb upstream of the TSS to 3 kb downstream of the TES in 50 bp bins. The genomic feature and peak densities around the TSSs region (–3, +3 k bp) were visualized by ChIPseeker (Yu et al, 2015). For annotation and further analysis of significant peaks, the de novo motif analysis was performed with the HOMER's findMotifsGenome. pl tool (Heinz et al, 2010) was used for motif analysis, with the region size were set as 200. The other plots were generated using R (v 3.4.2), and ggplot2 package available on Bioconductor. STRING online database (http://string-db.org) was used for analyzing the PPI of common genes and Cytoscape software (http://www.cytoscape.org/) was employed for visualizing the PPI network of common genes in Fig 4B.

### Nuclei of specific neurons chromatin immunoprecipitation

Newly eclosed flies were placed on standard fly food for 5 d under 29°C LD, and then the entrained flies were collected. Nuclei were obtained from 200–300 male heads using mini-INTACT protocol (Agrawal et al, 2019). Rabbit anti-GFP (G10362; Invitrogen) was used in mini-INTACT protocol. Then, ChIP for H3K27me3 on *zfh2*, *Vnd*, *mthl10*, and *Prosbeta2* genomic locus was performed using protocols described previously (Brind'Amour et al, 2015). Anti-trimethyl Histone H3 (Lys27) (ab6002; Abcam) was used in ChIP. All ChIP analyses were repeated three times with independent biological replicates. % Input was calculated by the following formula: % Input = $100 \times 2^{(-(Ct \, [ChIP] - (Ct \, [Input] - Log2 \, (Input \, Dilution \, Factor))))}$. The primer pairs used for qRT-PCR were listed in Table S6.

### Quantitative real-time PCR

Flies were collected on liquid nitrogen at indicated time points and stored at –80°C until all of the time points were collected (within 24h). Fly heads were then collected and homogenized with total RNA extraction reagent (#R401-01-AA; Vazyme) on ice using standard protocols for RNA. cDNA was generated with HiScript III All-in-one RT SuperMix Perfect kit (#R333; Vazyme), according to the manufacturer's protocol. Quantitative RT–PCR reaction was performed in the StepOne Real-Time PCR System (Applied Biosystems) using ChamQ SYBR qRT-PCR Master Mix (High ROX Premixed) (#Q341; Vazyme) with gene-specific primers.

Relative gene expression was calculated using the delta–delta Ct method with *Rp49* as internal control. The significance of differences between genotypes was tested by *t* test (GraphPad Prism). All experiments were done with at least three biological repeats, and sample size for each biological repeat was at least 20 fly heads, three technical repeats were done for each biological repeat. Primers used in this study for quantitative real-time PCR were listed in Table S6.

### Statistical analysis

Statistical analyses were carried out with Prism GraphPad, all data were expressed as the mean ± SEM. One-way ANOVA followed by Tukey post hoc test and unpaired *t* test were used to compare the differences between different genotypes. For all statistical tests, n.s. indicates no significant difference, *$P$ < 0.05, **$P$ < 0.01, ***$P$ < 0.001, and ****$P$ < 0.0001.

## Data Availability

TaDa sequencing data and experimental information have been deposited under NCBI and are available under PRJNA858184. The original contributions presented in the study are included in the article/Supplementary Material; further inquiries can be directed to the corresponding authors.

## Supplementary Information

## Acknowledgements

This work was supported by the National Natural Science Foundation of China (Grant No. 32070492 and 32122017) to J Du. We would like to thank Dr. Andrea H Brand (University of Cambridge, UK) for sharing the Dam-ID plasmids. We would like to thank EditSprings (https://www.editsprings.cn) for the expert linguistic services provided. We thank Dr. Jianming Zeng (University of Macau) and all the members of his bioinformatics team, Biotrainee, for generously sharing their experience and codes. The use of the biorstudio high-performance computing cluster at Biotrainee and The Shanghai HS Biotech Co., Ltd. for conducting the research reported in this article.

## Author Contributions

X Zhao: investigation, methodology, and writing—original draft, review, and editing.

X Yang: formal analysis, investigation, and writing—original draft, review, and editing.

P Lv: formal analysis, investigation, methodology, and writing—original draft, review, and editing.

Y Xu: software and formal analysis.

X Wang: software and supervision.

Z Zhao: validation and writing—review and editing.

J Du: conceptualization, supervision, funding acquisition, validation, and writing—original draft, review, and editing.

## Conflict of Interest Statement

The authors declare that they have no conflict of interest.

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
