## [Reviewer comments · Life Science Alliance]

Life Science Alliance

Polycomb regulates circadian rhythms in *Drosophila* in clock neurons

Xianguo Zhao, Xingzhuo Yang, Pengfei Lv, Yuetong Xu, Xiangfeng Wang, Zhangwu Zhao, and Juan Du

DOI: <https://doi.org/10.26508/lsa.202302140>

Corresponding author(s): Juan Du, China Agricultural University

Review Timeline:

Submission Date:	2023-05-08
Editorial Decision:	2023-06-08
Revision Received:	2023-08-28
Editorial Decision:	2023-09-22
Revision Received:	2023-10-06
Editorial Decision:	2023-10-11
Revision Received:	2023-10-15
Accepted:	2023-10-17

Scientific Editor: Novella Guidi

Transaction Report:

June 8, 2023

Re: Life Science Alliance manuscript #LSA-2023-02140

Dr. Juan Du
China Agricultural University
2# Yuanmingyuan West Rd
Beijing 100193
China

Dear Dr. Du,

Thank you for submitting your manuscript entitled "Polycomb regulates circadian rhythms in *Drosophila* in clock neurons" to Life Science Alliance. The manuscript was assessed by expert reviewers, whose comments are appended to this letter. We invite you to submit a revised manuscript addressing the Reviewer comments.

Thank you for this interesting contribution to Life Science Alliance. We are looking forward to receiving your revised manuscript.

Sincerely,

B. MANUSCRIPT ORGANIZATION AND FORMATTING:

Reviewer #1 (Comments to the Authors (Required)):

Zhao X. and colleagues have demonstrated a function for Polycomb protein Pc in the regulation of circadian rhythm within the *Drosophila* Clock neurons. The manuscript is well written, experiments are straightforward, the methodologies used are novel and cutting edge. However more details should be provided on the models chosen and some important and highly relevant controls are missing to warrant the publication of this work. Clarity should be implemented.

1. For this reviewer it was very difficult to approach all the genetic models presented in Figure 1 and their relative meaning. It would be advisable to find a simple, short and effective way to explain what the various models are for, in order to make the understanding of the experiments possible also for a wider audience. Methods should be substantiated. Abstract is not fully understandable.

Example of what the reviewer is referring to (but not limited to that):

Line 87-89: "In contrast to the w1118 and two control group flies including UAS-PcRNAi and tim-Gal4, both of which are 100% in percentage rhythmicity, 61.3% of the tim-Gal4/PcRNAi flies were rhythmic (Table 1)."

It is very hard to discriminate between proper controls and knock down flies. Also in the methodological part these information are missing. This reviewer asks for a proper description of the models chosen, and why they have been chosen.

2. The subsequent analyzes are well conducted and followable. However, to derive information about the specificity of the binding and more importantly, about the function of Pc in the four clusters of neurons, one important control is missing: "general neurons (line 150)".

Are the Pc binding sites identified in the four clusters absent in other neural cells that do not belong to the four clusters of neurons? How the Pc binding is modulated between clock and non clock neurons?

Ideally same experiments with TaDa should be performed in other neurons. However if this will be not possible, a validation on small scale with ChIP followed by qRT PCR may be of value. In this case would be interesting also to add ChIP for the histone modification.

3. How the regions bound by Pc have been identified?

Line 168-169: To identify Pc binding genes enriched in the above four clock neurons, expression data were analyzed from these clock neurons.

What does this mean? Why expression data have been included?

4. Be sure to cite all the panels presented and in a proper chronological order.

5. Be sure to perform statistics in all the panel reported (Ex. Figure 6 is missing)

6. Figure 6, the expression of Per and Tim in the validation with Pc KD are impossible to be evaluated by this reviewer...the genetic models used are incomprehensible.

Minor:

1. Bar plot can be substituted by box plot or barplot that display the dots in order to understand how data are distributed.

2. Figures will benefit by avoiding bold and italic font.

3. Tables should be uniform for their esthetics.

4. Line 84-85: "In the constant darkness (DD) conditions the tim-85 Gal4/PcRNAi flies showed a severely disrupted circadian rhythm (Table 1)." How it has been measured?

Reviewer #2 (Comments to the Authors (Required)):

1. A short summary of the paper, including description of the advance offered to the field.

In this paper, the authors investigate the role of Pc in circadian rhythms in *Drosophila*. Firstly, they show that knockdown of Pc in clock neurons causes a disruption to circadian rhythms. Then the paper centres on their TaDa-seq data - whereby they map the binding of Pc across the genome in different classes of neurons (l-LNvs, LNds, DN1s, s-LNvs), using DamID where the expression of Dam is driven in a targeted way. They analyse this data and show that Pc bind to promoters, and that target sites vary in the different classes of neuron. They identify the genes that are bound by Pc, the pathways/ontologies, and find significant overlap with known circadian rhythm genes. They show their TaDa profiles in the different neuron classes across known circadian genes, showing more prominent binding in l-LNvs and s-LNvs generally. They then validate their findings in the well characterised *per* and *tim* circadian genes, showing DamID qPCR binding in the different neuron classes, and the changes in expression that accompany Pc knockdown in the different neurons. Although the paper lacks mechanistic insight into how Pc is regulating the genes, the involvement of Pc in circadian rhythms is novel to my knowledge, and the TaDa data in the different neurons is of value to the field.

2. For each main point of the paper, please indicate if the data are strongly supportive. If not, explicitly state the additional experiments essential to support the claims made and the timeframe that these would require.

Pc gene expression is required for behavioral circadian rhythms (Figure 1). Data are strongly supportive.

Global features and functional classification of Pc bound genes identified by TaDa (Figure 2 and 3). Data are supportive. However, explanation or re-plotting is required for 2G, where the Dam signal is higher than the Dam-PC signal in all clusters. It would be interesting to analyse the published RNA-seq data that the authors cite and see where the Pc bound genes fall in terms of expression levels i.e. is Pc binding to active or inactive genes. This would support the discussion of Figure 6 data (see below).

The TaDa analysis shows that Pc binds to the genes involved in the circadian rhythm and rhythmic process pathways (Figure 4 and 5). Data are supportive. The group of 5045 circadian rhythm related genes is very large though - maybe there is a more refined published gene list that could be compared to the PC-bound gene lists.

There is a direct role for Pc in regulating circadian rhythms through specific clock genes (Figure 6). This requires some additional data to make the claim.

- the expression qRT-PCR analysis in Fig. 6 L-S should be carried out in samples from C929-Gal4-marked neurons since there is prominent binding of Pc to *per* and *tim* in this sample.

- To make the conclusion that "In addition, we checked the Pc binding on *Per* locus at four time points across 24 hours. The results showed that the Pc binding was not statistically oscillated (Figure S7)" the analysis should be conducted in C929-Gal4 marked neurons with primer set 1, where the strongest binding of Pc to *per* is observed (Fig. 6B).

- The authors should discuss more thoroughly the results in Fig 6 L-S

a) why they think they see the phenotype in Fig 6L-S, in which *tim*-Gal neurons show upregulation of *per* and *tim* in Pc knockdown, whereas other classes of clock neurons show downregulation in Pc knockdown.

b) Pc is commonly a repressor, so why do they think they see downregulation?

c) Why do they see downregulation in Pc knockdown of genes where there is no prominent Pc binding e.g. *tim* and *per* in R18H11-Gal4 marked neurons

3. Lastly, indicate any additional issues you feel should be addressed (text changes, data presentation, statistics etc.).

None

This study shows that the epigenetic regulator *Polycomb (Pc)* regulates the circadian rhythm by binding to the genes involved in the circadian rhythm in the *Drosophila* clock neurons.

Reviewer #1 (Comments to the Authors (Required)):

Zhao X. and colleagues have demonstrated a function for Polycomb protein Pc in the

regulation of circadian rhythm within the *Drosophila* Clock neurons. The manuscript is well written, experiments are straightforward, the methodologies used are novel and cutting edge. However more details should be provided on the models chosen and some important and highly relevant controls are missing to warrant the publication of this work. Clarity should be implemented.

1. For this reviewer it was very difficult to approach all the genetic models presented in Figure 1 and their relative meaning. It would be advisable to find a simple, short and effective way to explain what the various models are for, in order to make the understanding of the experiments possible also for a wider audience. Methods should be substantiated. Abstract is not fully understandable.

Example of what the reviewer is referring to (but not limited to that):

Line 87-89: "In contrast to the w1118 and two control group flies including UAS-PcRNAi and tim-Gal4, both of which are 100% in percentage rhythmicity, 61.3% of the tim-Gal4/PcRNAi flies were rhythmic (Table 1)."

It is very hard to discriminate between proper controls and knock down flies. Also in the methodological part these information are missing. This reviewer asks for a proper description of the models chosen, and why they have been chosen.

Answer: We apologize for the confusions. To distinguish the genotypes for the knockdown flies, we have labeled them in blue in Figure 1. In the revised table 1, three types of genotypes were

indicated, including Control phenotypes, *Pc* down regulation phenotypes, *Pc* up regulation phenotypes. In the method session, the corresponding descriptions were added to the last paragraph of the “fly stocks” session, including description of the models chosen, and why they have been chosen.

“In this study, downregulation of Polycomb (*Pc*) levels by *Pc* RNAi in *Drosophila* clock neurons (*tim-Gal4* marked most of the clock neurons) resulted in abnormal circadian rhythms. To further elucidate which clock neurons are required for the *Pc* gene to regulate the circadian rhythm, we selected several Gal4s specifically expresses in key clusters of clock neurons to drive the *Pc* RNAi. These drivers include *C929-Gal4* (marked clusters including l-LNvs), *R6-Gal4* (marked clusters including s-LNvs), *R18H11-Gal4* (marked clusters including DN1) and *DVpdf-Gal4*, *pdf-Gal80* (marked clusters including LNds). Based on the purpose of our experiments, we checked the phenotypes of flies with altered *Pc* levels in LD/DD conditions.”

2. The subsequent analyzes are well conducted and followable. However, to derive information about the specificity of the binding and more importantly, about the function of *Pc* in the four clusters of neurons, one important control is missing: "general neurons (line 150)".

Answer: We apologize for the inaccuracy of this statement. The statement “the general neuronal binding genes” in line 150 should be “the general background bindings”. This has been revised in the updated version. The experimental design of TaDa assay in this experiment is to identify *Pc* binding in the four clusters of neurons, which was achieved by expressing Dam-*Pc* specifically in certain neuron clusters. The general background bindings were eliminated by including of Dam lines (eg. *C929-Gal4/tub-Gal80ts;UAS-Dam/+*, *et al.*), which were clarified in the last paragraph of the “TaDa sample preparation and data analysis” session of the material and methods.

Are the *Pc* binding sites identified in the four clusters absent in other neural cells that do not belong to the four clusters of neurons? How the *Pc* binding is modulated between clock and non clock neurons?

Ideally same experiments with TaDa should be performed in other neurons. However if this will be not possible, a validation on small scale with CHIP followed by qRT PCR may be of value. In this case would be interesting also to add CHIP for the histone modification.

Answer: I agree. Exploring the *Pc* binding profile in neurons other than clock neurons is indeed an

intriguing question. To gain insight into *Pc* binding differences between clock neurons and non-clock neurons, we conducted TaDa assays followed by PCR to assess *Pc* bindings at specific gene loci. Experiments were done using the following genotypes:

Tub-G80^{ts};Dam>TH-Gal4

Tub-G80^{ts};DamPc>TH-Gal4

Tub-G80^{ts};Dam>tim-Gal4

Tub-G80^{ts};DamPc>tim-Gal4

To verify the histone modification on these sites, we also performed tissue specific ChIP for the histone modification by using the UAS-unc84GFP labeling and isolation of nuclei of specific neuron clusters. Experiments were done using the following genotypes:

Tub-G80^{ts}/UAS-unc84GFP;TH-Gal4/+

Tub-G80^{ts}/tim-Gal4;UAS-unc84GFP/+

The result has been put into Figure 7.

3. How the regions bound by *Pc* have been identified?

Line 168-169: To identify *Pc* binding genes enriched in the above four clock neurons, expression data were analyzed from these clock neurons.

What does this mean? Why expression data have been included?

Answer: We apologize for the inaccuracy of this statement. Thank you for bringing this to our attention. This sentence should be “To identify *Pc* binding genes enriched in the above four clock neurons, TaDa data was analyzed from these clock neurons.” This has been revised in the updated version.

4. Be sure to cite all the panels presented and in a proper chronological order.

Answer: In the updated version, we have thoroughly revised the manuscript and figures. We have also ensured that all the panels are properly cited in a chronological order. (eg. Figure S1).

5. Be sure to perform statistics in all the panel reported (Ex.Figure 6 is missing)

Answer: Thank you for the suggestion. We have performed the statistical analysis as recommended by the reviewer, and the results have been incorporated into the updated Figure 6.

6. Figure 6, the expression of *Per* and *Tim* in the validation with *Pc* KD are impossible to be evaluated by this reviewer...the genetic models used are incomprehensible.

Answer: The inclusion of the *tub-Gal80^{ts}* was to activate the *Gal4* in a temporally specific manner and preventing the early expression-associated lethality (McGuire et al., 2003). *Gal80*'s presence has an inhibitory effect on *Gal4* activity. The inclusion of *Gal80^{ts}* enables *Gal80* be deactivated upon the high temperature treatment, which activates the *Gal4* in a temporally specific manner. In

Figure 6 L-U, the red line represents genotypes involving temporal and spatial RNAi of *Pc*, while the black line corresponds to the control group.

Ref:

McGuire SE, Le PT, Osborn AJ, Matsumoto K, Davis RL. Spatiotemporal rescue of memory dysfunction in *Drosophila*. *Science*. 2003 Dec 5;302(5651):1765-8. doi: 10.1126/science.1089035.

Minor:

1. Bar plot can be substituted by box plot or barplot that display the dots in order to understand how data are distributed.

Answer: As suggested by the reviewer, bar plots have been substituted by barplot that display the dots. The results have been incorporated into the updated figures, including Figure 1G-I, Figure 6B-K, Figure 7, Figure S1M-O, Figure S2 and Figure S7B-G.

2. Figures will benefit by avoiding bold and italic font.

Answer: Thank you for the suggestion. As suggested by the reviewer, we have eliminated the use of bold fonts entirely. Italic font was also avoided except when it is necessary, as italic fonts are typically employed for gene names and genotypes in the *Drosophila* research field.

3. Tables should be uniform for their esthetics.

Answer: As suggested by the reviewer, all main Tables and supplementary Tables are uniform.

4. Line 84-85: "In the constant darkness (DD) conditions the *tim-85 Gal4/PcRNAi* flies showed a severely disrupted circadian rhythm (Table 1)." How it has been measured?

Answer: According to the standard method for measuring circadian rhythm of *Drosophila*, *Drosophila* Activity Monitoring (DAM) system (Trikinetics, MA, U.S.) were used in this study. Briefly, individual fly is placed in tube where their movement was continuously monitored using a simple and robust array of infrared beams, recording movements per second.. The resulting data collected over periods of days is uploaded periodically to a computer for subsequent analysis. A diagram has been included below to better illustrate this mechanism (As show at <https://trikinetics.com/>). A detailed description of the method can be found in the material and method session under the title “*Drosophila* Activity Monitor-based method for circadian rhythm measurement”.

©2000 Scientific American Vol. 282 No 3 THE TICK-TOCK OF THE BIOLOGICAL CLOCK, Michael W. Young

Reviewer #2 (Comments to the Authors (Required)):

1. A short summary of the paper, including description of the advance offered to the field.

In this paper, the authors investigate the role of Pc in circadian rhythms in *Drosophila*. Firstly, they show that knockdown of Pc in clock neurons causes a disruption to circadian rhythms. Then the paper centres on their TaDa-seq data - whereby they map the binding of Pc across the genome in different classes of neurons (l-LNvs, LNds, DN1s, s-LNvs), using DamID where the expression of Dam is driven in a targeted way. They analyse this data and show that Pc bind to promoters, and that target sites vary in the different classes of neuron. They identify the genes that are bound by Pc, the pathways/ontologies, and find significant overlap with known circadian rhythm genes. They show their TaDa profiles in the different neuron classes across known circadian genes, showing more prominent binding in l-LNvs and s-LNvs generally. They then validate their findings in the well characterised *per* and *tim* circadian genes, showing DamID qPCR binding in the different

neuron classes, and the changes in expression that accompany Pc knockdown in the different neurons. Although the paper lacks mechanistic insight into how Pc is regulating the genes, the involvement of Pc in circadian rhythms is novel to my knowledge, and the TaDa data in the different neurons is of value to the field.

2. For each main point of the paper, please indicate if the data are strongly supportive. If not, explicitly state the additional experiments essential to support the claims made and the timeframe that these would require.

Pc gene expression is required for behavioral circadian rhythms (Figure 1). Data are strongly supportive.

Global features and functional classification of Pc bound genes identified by TaDa (Figure 2 and 3). Data are supportive. However, explanation or re-plotting is required for 2G, where the Dam signal is higher than the Dam-PC signal in all clusters. It would be interesting to analyse the published RNA-seq data that the authors cite and see where the Pc bound genes fall in terms of expression levels i.e. is Pc binding to active or inactive genes. This would support the discussion of Figure 6 data (see below).

Answer: Thanks for pointing this out. Yes, all analysis conducted was compared with the Dam controls to eliminate non-specific effects (See material and method session for details). Figure 2F showed values for each sample, while Figure 2G showed the final results compared with the Dam controls. All the other analysis were done by eliminating the background effects of Dam.

The Dam signal was higher than the Dam-Pc signal in Figure 2F troubled us for a while. However, when we look back into the literature, we found a paper from Nature communication has data got from Dam-Pc (Marshall et al., 2017). We analyzed their data using the same procedure and got the similar result (Figure showed below, comparing with figure 2F). We believe that the reason behind this phenomenon may be attributed the weak toxicity of Dam-Pc expression to the cells. This potential toxicity could have influenced the cell cycle or cell survival and resulted in a much less cell number in the Dam-Pc samples than the Dam only samples.

Consequently, this could explain the systematically higher signal in the Dam control samples. However, after comparing with the Dam only signals, the relative signals from the Dam-*Pc* samples between genes were not affected by the systematically higher Dam signals. So, the data we got were valid.

Ref :

Marshall OJ, Brand AH. Chromatin state changes during neural development revealed by in vivo cell-type specific profiling. *Nat Commun.* 2017 Dec 22;8(1):2271. doi: 10.1038/s41467-017-02385-4.

Thank you very much for the valuable suggestion. We have analyzed the *Pc* RNAi data from the dataset GSE100547 (<https://www.ncbi.nlm.nih.gov/geo/query/acc.cgi?acc=GSE100547>)^[1] and GSE157793 (<https://www.ncbi.nlm.nih.gov/geo/query/acc.cgi?acc=GSE157793>)^[2] and observed a significant decrease in the expression levels of *per* and *tim* genes (as shown in the figure below). This finding further supports the notion that *Pc* protein plays a positive regulatory role in the circadian rhythm control of *per*, which showed a significant change in dataset GSE157793. It provides a deeper understanding of the involvement of *Pc* protein in circadian rhythm regulation.

1. Pherson M, Misulovin Z, Gause M, Mihindikulasuriya K et al. Polycomb repressive complex 1 modifies transcription of active genes. *Sci Adv* 2017 Aug;3(8):e1700944. PMID: 28782042
2. Tauc HM, Rodriguez-Fernandez IA, Hackney JA, Pawlak M et al. Age-related changes in

polycomb gene regulation disrupt lineage fidelity in intestinal stem cells. *Elife* 2021 Mar 16;10. PMID: 33724181

The TaDa analysis shows that Pc binds to the genes involved in the circadian rhythm and rhythmic process pathways (Figure 4 and 5). Data are supportive. The group of 5045 circadian rhythm related genes is very large though - maybe there is a more refined published gene list that could be compared to the PC-bound gene lists.

Answer: The results of the separate comparisons for the 5045 circadian rhythm-related genes are shown in the figure below. This additional analysis allows for a more detailed examination of the binding patterns of Pc protein to these genes and provides further support for our findings.

There is a direct role for Pc in regulating circadian rhythms through specific clock genes (Figure 6). This requires some additional data to make the claim.

- the expression qRT-PCR analysis in Fig. 6 L-S should be carried out in samples from C929-Gal4-marked neurons since there is prominent binding of Pc to per and tim in this sample.

Answer: As suggested by the reviewer, RT-qPCR carried out in samples from *C929-Gal4* marked neurons revealed that the *Per* and *Tim* expression were down regulated by *Pc* knocking down in

the *C929-Gal4* marked neurons (Figure 6P, U). This result aligns with the trends observed in the down-regulation of *Pc* in the *pdf-Gal4* marked neurons. These results have been added in the updated figure 6P, U.

- To make the conclusion that "In addition, we checked the *Pc* binding on *Per* locus at four time points across 24 hours. The results showed that the *Pc* binding was not statistically oscillated (Figure S7)" the analysis should be conducted in *C929-Gal* marked neurons with primer set 1, where the strongest binding of *Pc* to *per* is observed (Fig. 6B).

Answer: As suggested by the reviewer, we conducted TaDa-qPCR with *Per* primer set 1 at four time points across 24 hours in *C929-Gal4* marked neurons. We found that *Pc* binding *Per* primer set 1 was statistically oscillated (Figure S7G). These results have been included in the revised manuscript and updated Figure S7G.

- The authors should discuss more thoroughly the results in Fig 6 L-S

a) why they think they see the phenotype in Fig 6L-S, in which *tim-Gal* neurons show upregulation of *per* and *tim* in *Pc* knockdown, whereas other classes of clock neurons show downregulation in *Pc* knockdown.

b) *Pc* is commonly a repressor, so why do they think they see downregulation?

c) Why do they see downregulation in *Pc* knockdown of genes where there is no prominent *Pc* binding e.g. *tim* and *per* in *R18H11-Gal4* marked neurons

Answer: Thank you for pointing out this. I agree that we should have been discussed more thoroughly about it. One paragraph was added to the discussion section.

"The relationship between expression of the *Pc* downstream targets and *Pc* bound targets were complex. *Pc* has direct downstream targets (bound by *Pc*) and indirect downstream targets (not bound by *Pc*). As shown in figure 6L-U, we checked the expression level of clock genes. *Pc* can function as both repressor or activator (Morey L et al., 2010; Lv X et al., 2017). Moreover, as discussed above, in some cases, non-clock neurons were targeted in this study. So, when *Pc* is knockdown, its direct downstream target genes can be either activated or repressed under different conditions in this study. That's why we see either up regulation or down regulation of *Pc* targets.

Pc could also have indirect target, whose expression could be regulated by direct downstream targets of *Pc*. This could explain what we see in *tim* and *per* in *R18H11-Gal4* marked neurons.”

3. Lastly, indicate any additional issues you feel should be addressed (text changes, data presentation, statistics etc.).

None

September 22, 2023

Re: Life Science Alliance manuscript #LSA-2023-02140R

Dr. Juan Du
China Agricultural University
2# Yuanmingyuan West Rd
Beijing 100193
China

Dear Dr. Du,

Thank you for submitting your revised manuscript entitled "Polycomb regulates circadian rhythms in *Drosophila* in clock neurons" to Life Science Alliance. The manuscript has been seen by the original reviewers whose comments are appended below. While the reviewers continue to be overall positive about the work in terms of its suitability for Life Science Alliance, some important issues remain.

Our general policy is that papers are considered through only one revision cycle; however, given that the suggested changes are relatively minor, we are open to one additional short round of revision. Please note that I will expect to make a final decision without additional reviewer input upon re-submission.

Please submit the final revision within one month, along with a letter that includes a point by point response to the remaining reviewer comments.

To upload the revised version of your manuscript, please log in to your account: <https://lsa.msubmit.net/cgi-bin/main.plex>
You will be guided to complete the submission of your revised manuscript and to fill in all necessary information.

B. MANUSCRIPT ORGANIZATION AND FORMATTING:

Sincerely,

Reviewer #1 (Comments to the Authors (Required)):

Unfortunately, this reviewer finds the current form of the work still very difficult to be read.

In the previous review, this reviewer asks to improve the clarity: numerous suggestions were made and the authors were

strongly urged to improve the way the data were presented, to more easily clarify for example the genetic models used for a broad audience. This reviewer does not feel that the work has been much improved in this regard. Even genomic traces are difficult to interpret and of poor graphical quality, the comparisons to be made are not immediate. This reviewer believes that, to be published, this work must benefit from a global reorganization that considers the good of the reader, the readability of the article, the clarity in the presentation of the data as the first priority.

Reviewer #2 (Comments to the Authors (Required)):

The manuscript is improved from the initial submission, with some extra supportive data added, and improved clarity and figures (particularly the graph presentation).

From my comments in the first round of revision, most have been discussed and/or additional data added to the paper. However, the graphs provided in the rebuttal in response to my query: "The group of 5045 circadian rhythm related genes is very large though - maybe there is a more refined published gene list that could be compared to the PC-bound gene lists" needs a little more information. How many genes are in each of the groups they examine, and where do those lists come from? Is the percentage circadian-related Pc bound (TaDa data) targets similar in each cell type for the different groups of circadian gene lists? Could this be provided in Supplementary data for all readers?

My other remaining concern is that no negative control region is provided for the TaDa-qPCR (Fig 6) or for the H3K27me3 ChIP-qPCR (Fig 7), and so it is impossible to say whether or not there is binding e.g. are levels that are below those at *iab-7* still sig above background?

Minor comments:

- Define acronyms at first use e.g. PDF, s-LNvs, l-LNvs, and LN_d
- Figure citations incorrect for statement line 304 "While, in DVpdf-Gal4, pdf-Gal80 marked neurons, Pc had binding comparable with *iab-7* on Tim but not on Per gene locus (Figure 6E, J). In R18H11-Gal4 marked neurons, Pc had much less binding on both Per and Tim gene locus (Figure 6F, K)." - these are the wrong way round

Reviewer #1 (Comments to the Authors (Required)):

Unfortunately, this reviewer finds the current form of the work still very difficult to be read.

In the previous review, this reviewer asks to improve the clarity: numerous suggestions were made and the authors were strongly urged to improve the way the data were presented, to more easily clarify for example the genetic models used for a broad audience. This reviewer does not feel that the work has been much improved in this regard. Even genomic traces are difficult to interpret and of poor graphical quality, the comparisons to be made are not immediate.

This reviewer believes that, to be published, this work must benefit from a global reorganization that considers the good of the reader, the readability of the article, the clarity in the presentation of the data as the first priority.

Answer: We apologize for any confusion caused. To provide clearer clarification of the genetic models used, we have made further improvements to the data. In Figure 1 and Figure S1, we have now labeled the meaning of the genetic models. Additionally, in the method session (Last paragraph under the title: *Drosophila* Activity Monitor-based method for circadian rhythm measurement), we have included descriptions of the parameters, namely Activity, Power, periods, and rhythmicity.

“For the circadian analysis, activity counts for each fly were binned every 30 minutes. Power represents the relative strength of the rhythm during DD. Flies with a power value greater than or equal to 10, along with a "width" value of 1.5 or more (indicating the number of peaks above the periodogram 95% confidence line in 30-minute increments), were considered rhythmic. Similarly, for DD rhythmicity, rhythmic flies were defined as those with a minimum period peak power of 10 or above. Period calculations took into account all flies with power-significance values greater than or equal to 10. Periods were calculated for each individual fly using X2 periodogram analysis and then pooled to obtain a group average for each genotype.”

Regarding graphical quality, we have provided figures in a higher quality TIFF format. These figures have been revised in the updated version.

Reviewer #2 (Comments to the Authors (Required)):

The manuscript is improved from the initial submission, with some extra supportive data added, and improved clarity and figures (particularly the graph presentation).

From my comments in the first round of revision, most have been discussed and/or additional data added to the paper. However, the graphs provided in the rebuttal in response to my query: "The group of 5045 circadian rhythm related genes is very large though - maybe there is a more refined published gene list that could be compared to the PC-bound gene lists" needs a little more information. How many genes are in each of the groups they examine, and where do those lists come from? Is the percentage circadian-related Pc bound (TaDa data) targets similar in each cell type for the different groups of circadian gene lists? Could this be provided in Supplementary data for all readers?

Answer: We appreciate your request for more information. Here are the details for each gene group, including the number of genes and their sources:

1. Clock Binding Gene List: This list comprises 1517 genes and is sourced from a study that extensively investigated the role of CLK as a master regulator of the circadian clock in *Drosophila*, identifying its target genes and their regulation (Abruzzi et al., 2011).

2. Clock Neuro List: It consists of 249 genes and is derived from research that focused on identifying specific genes and their roles in neuronal subgroups of the *Drosophila* circadian circuit, shedding light on the molecular mechanisms underlying circadian rhythms (Nagoshi et al., 2010).

3. CGDB list (Circadian Gene Database): This database contains over 73,000 validated circadian genes across various organisms, including animals, plants, and fungi. It provides extensive information on oscillations, tissue specificity, and potential associations with diseases (Li et al., 2017).

4. JTK Cycle gene list: This dataset encompasses 676 genes obtained from a study involving transcriptome sequencing on a large number of samples from wild-type *Drosophila* at multiple time points with different circadian periods. We used the JTK Cycle software to screen for rhythmically correlated genes and combined results from both sexes to obtain this gene list

(Kumar, S., 2021).

5. *Per*⁰¹ mutant gene list: Comprising 1475 genes, this list originates from a study that conducted mutant RNA sequencing analysis of *Drosophila melanogaster* brains, revealing many genes with 24-hour rhythmic expression patterns under different light conditions (Hughes et al., 2012).

We conducted a reevaluation of the proportion of Pc binding genes within each of the gene sets. In each cell type, the percentage of Pc-bound targets associated with circadian rhythms appeared to be similar (Figure S4C-D).

Finally, we have already re-described these results in the main text, specifically in lines 218-226 and lines 236-240 (First paragraph under the title: Identification of *Pc* genomic binding sites in different clusters of clock neurons). Corresponding figures have also been incorporated into Supplementary Figure S4C-D.

References:

- Abruzzi, K. C., Rodriguez, J., Menet, J. S., Desrochers, J., Zadina, A., Luo, W., et al. (2011). *Drosophila* CLOCK target gene characterization: implications for circadian tissue-specific gene expression. *Gene Dev.* 25, 2374-2386. doi.org/10.1101/gad.178079.111
- Nagoshi, E., Sugino, K., Kula, E., Okazaki, E., Tachibana, T., Nelson, S., et al. (2010). Dissecting differential gene expression within the circadian neuronal circuit of *Drosophila*. *Nat Neurosci.* 13, 60-68. doi.org/10.1038/nn.2451
- Li, S., Shui, K., Zhang, Y., Lv, Y., Deng, W., Ullah, S., et al. (2017). CGDB: a database of circadian genes in eukaryotes. *Nucleic Acids Res.* 45, D397-D403. doi.org/10.1093/nar/gkw1028
- Kumar, S., Tunc, I., Tansey, T. R., Pirooznia, M., and Harbison, S. T. (2021). Identification of Genes Contributing to a Long Circadian Period in *Drosophila Melanogaster*. *J Biol Rhythm.* 36, 239-253. doi.org/10.1177/0748730420975946
- Hughes, M. E., Grant, G. R., Paquin, C., Qian, J., and Nitabach, M. N. (2012). Deep sequencing the circadian and diurnal transcriptome of *Drosophila* brain. *Genome Res.* 22, 1266-1281. doi.org/10.1101/gr.128876.111

My other remaining concern is that no negative control region is provided for the TaDa-qPCR (Fig 6) or for the H3K27me3 ChIP-qPCR (Fig 7), and so it is impossible to say whether or not there is binding e.g. are levels that are below those at *iab-7* still sig above background?

Answer: Thank you for your valuable suggestion. As recommended by the reviewer, we have incorporated a negative control PGRP-LE gene into the TaDa-qPCR (Fig 6) and the H3K27me3 ChIP-qPCR (Fig 7). The inclusion of PGRP-LE as a negative control allows us to assess non-PcG target activity (Du et al., 2016). These results have been included in the updated figures.

Reference:

Du, J., Zhang, J., He, T., Li, Y., Su, Y., Tie, F., et al. (2016). Stuxnet Facilitates the Degradation of Polycomb Protein during Development. *Dev Cell*. 37, 507-519. doi.org/10.1016/j.devcel.2016.05.013

Minor comments:

- Define acronyms at first use e.g. PDF, s-LNvs, l-LNvs, and LNd

Answer: As recommended by the reviewer, we defined acronyms upon their initial usage, e.g. s-LNvs (small ventral lateral neurons), l-LNvs (large ventral-lateral neurons), LNds (dorsal-lateral clock neurons), DN1s (Dorsal neurons 1), DN2s (Dorsal neurons 2), DN3s (Dorsal neurons 3), PDF (Pigment-Dispersing Factor). The updated text (lines 30-38) now includes the addition of these acronyms.

- Figure citations incorrect for statement line 304 "While, in DVpdf-Gal4, pdf-Gal80 marked neurons, Pc had binding comparable with iab-7 on Tim but not on Per gene locus (Figure 6E, J). In R18H11-Gal4 marked neurons, Pc had much less binding on both Per and Tim gene locus (Figure 6F, K)." - these are the wrong way round

Answer: We sincerely apologize for the inaccuracy of this statement. This has been revised in the updated version.

October 11, 2023

RE: Life Science Alliance Manuscript #LSA-2023-02140RR

Dr. Juan Du
China Agricultural University
2# Yuanmingyuan West Rd
Beijing 100193
China

Dear Dr. Du,

Thank you for submitting your revised manuscript entitled "Polycomb regulates circadian rhythms in Drosophila in clock neurons". We would be happy to publish your paper in Life Science Alliance pending final revisions necessary to meet our formatting guidelines.

- please add the Twitter handle of your host institute/organization as well as your own or/and one of the authors in our system
- please make sure the author order in your manuscript and our system match
- please upload your Tables in editable .doc or Excel format

A. FINAL FILES:

B. MANUSCRIPT ORGANIZATION AND FORMATTING:

****It is Life Science Alliance policy that if requested, original data images must be made available to the editors. Failure to provide**

original images upon request will result in unavoidable delays in publication. Please ensure that you have access to all original data images prior to final submission.**

The license to publish form must be signed before your manuscript can be sent to production. A link to the electronic license to publish form will be sent to the corresponding author only. Please take a moment to check your funder requirements.

Sincerely,

October 17, 2023

RE: Life Science Alliance Manuscript #LSA-2023-02140RRR

Dr. Juan Du
China Agricultural University
2# Yuanmingyuan West Rd
Beijing 100193
China

Dear Dr. Du,

Thank you for submitting your Research Article entitled "Polycomb regulates circadian rhythms in *Drosophila* in clock neurons". It is a pleasure to let you know that your manuscript is now accepted for publication in Life Science Alliance. Congratulations on this interesting work.

DISTRIBUTION OF MATERIALS:

Again, congratulations on a very nice paper. I hope you found the review process to be constructive and are pleased with how the manuscript was handled editorially. We look forward to future exciting submissions from your lab.

Sincerely,
